# Primal-Dual Graph Neural Networks for General NP-Hard Combinatorial Optimization

## Abstract

*Neural algorithmic reasoning (NAR)* seeks to train neural networks, particularly Graph Neural Networks (GNNs), to simulate and generalize traditional algorithms, enabling them to perform structured reasoning on complex data. Previous research has primarily focused on algorithms for polynomial-time-solvable problems. However, many of the most critical problems in practice are NP-hard, exposing a critical gap in NAR. In this work, we propose a general NAR framework to learn algorithms for NP-hard problems, built on the classical primal-dual framework for designing efficient approximation algorithms. We enhance this framework by integrating optimal solutions to these NP-hard problems, enabling the model to surpass the performance of the approximation algorithms it was initially trained on. To the best of our knowledge, this is the first NAR method explicitly designed to surpass the performance of the classical algorithm on which it is trained. We evaluate our framework on several NP-hard problems, demonstrating its ability to generalize to larger and out-of-distribution graph families. In addition, we demonstrate the practical utility of the framework in two key applications: as a warm start for commercial solvers to reduce search time, and as a tool to generate embeddings that enhance predictive performance on real-world datasets. Our results highlight the scalability and effectiveness of the NAR framework for tackling complex combinatorial optimization problems, advancing their utility beyond the scope of traditional polynomial-time-solvable problems.

## 1 Introduction

The growing interest in teaching neural networks to reason algorithmically stems from the potential to combine the power of deep learning with the structured logic of classical algorithms. Classical algorithms offer step-by-step reasoning that guarantees correctness and interpretability, while neural networks excel at learning directly from data and handling unstructured, real-world inputs. The intersection of these fields has given rise to the emerging domain of Neural Algorithmic Reasoning (NAR) (Veličković & Blundell, 2021). NAR aims to train neural networks to replicate the operations of traditional algorithms, such as Bellman-Ford for the shortest-path problem. By embedding algorithmic knowledge into models, NAR not only enhances generalization to unseen problem instances, but also enables direct handling of real-world data, bypassing the need for extensive feature engineering to adapt problems to abstract algorithmic formats. For instance, a model pre-trained with Bellman-Ford knowledge can be applied to tackle real-world routing and transportation problems. This flexibility allows models to integrate rich, domain-specific features—such as weather conditions or network congestion—into their solutions.

Early research has shown promising results for NAR, particularly using Graph Neural Networks (GNNs) (Xu et al., 2020; Veličković et al., 2020; Ibarz et al., 2022; Deac et al., 2021; Xhonneux et al., 2021; Numeroso et al., 2023; Bevilacqua et al., 2023; Rodionov & Prokhorenkova, 2023). However, much of the current work has focused on algorithms for polynomial-time-solvable problems, in particular the 30 classic algorithms (e.g., sorting, search, graph) provided by the CLRS-30 benchmark (Veličković et al., 2022). This creates a significant gap when applying NAR to real-world problems, many of which are inherently NP-hard. This gap is critical, as the motivation behind NAR is to enable the transfer of algorithmic knowledge to tackle complex, real-world datasets effectively.

We propose **Primal-Dual Graph Neural Networks (PDGNNs)**, a general NAR framework to learn algorithms for NP-hard problems, significantly extending its applicability beyond the polynomial-time-solvable regime. Our approach builds on the classical primal-dual framework (Goemans & Williamson, 1996), a widely used technique for designing efficient approximation algorithms for NP-hard problems. As we show, the iterative nature of primal-dual algorithms naturally aligns with the message-passing operations of GNNs, making them an ideal foundation for our method. We provide theoretical guarantees showing that PDGNN can exactly replicate the classical primal-dual algorithm, establishing a strong theoretical basis for our framework.

Additionally, by incorporating supervision signals from optimal solutions of small problem instances, we enable PDGNN to surpass the performance of traditional primal-dual methods.[1] Unlike prior works that focus solely on replicating algorithmic behavior, our framework is explicitly designed to outperform the algorithm it is trained on.

We validate our framework on the minimum vertex cover, minimum set cover, and minimum hitting set problems. Empirically, we show that models trained on small graphs generalize effectively, outperforming approximation algorithms on larger instances and unseen graph families. Our results show that GNNs can acquire algorithmic reasoning from approximation algorithms for NP-hard problems, enabling robust generalization to unseen instances.

We further highlight the practical utility of PDGNN in two real-world scenarios. First, we use PDGNN outputs to warm start a commercial integer programming solver, significantly reducing both its search time and the size of its search trees. Second, we deploy the pretrained PDGNN on real-world datasets where the vertex cover problem is closely related to a classification task, achieving improved performance using the learned embeddings. These real-world applications underscore the significance of extending NAR to the NP-hard domain, breaking the constraints of polynomial-time-solvable problems and paving the way for solving more complex and impactful real-world challenges.

## 2 RELATED WORKS

**Neural algorithmic reasoning (NAR)**   The algorithmic alignment framework proposed by Xu et al. (2020) suggests that GNNs are particularly well-suited for learning dynamic programming algorithms due to their shared aggregate-update mechanism. Additionally, Veličković et al. (2020) demonstrates the effectiveness of GNNs in learning graph algorithms such as BFS and Bellman-Ford. These foundational works have contributed to the development of NAR (Veličković & Blundell, 2021), which investigates the potential of neural networks, particularly GNNs, to simulate traditional algorithms. However, much of the current work has focused on algorithms for polynomial-time-solvable problems (Ibarz et al., 2022; Bevilacqua et al., 2023; Rodionov & Prokhorenkova, 2023), in particular the 30 classic algorithms provided by the CLRS benchmark (Veličković et al., 2022), leaving NP-hard problems largely unexplored in NAR. This reveals a major limitation in extending NAR to real-world problems, where many challenges are inherently NP-hard. Overcoming this limitation is essential, as NAR aims to transfer algorithmic knowledge to handle complex and practical datasets effectively.

**NAR for NP-hard problems**   The most closely related work attempting to extend NAR to NP-hard problems is Georgiev et al. (2023c). Their approach involves pretraining a GNN on algorithms for polynomial-time-solvable problems (e.g., Prim's algorithm for MST) and using transfer learning to address NP-hard problems (e.g., TSP). However, this method is not general, as it requires carefully selecting a specific algorithm for pretraining and identifying a meaningful connection between the polynomial-time problem and the target NP-hard problem. In contrast, our method is inherently general, leveraging the primal-dual algorithm—a versatile framework applicable to a wide range of NP-hard problems. Additionally, our approach directly learns an approximation algorithm tailored to NP-hard problems, tackling unique challenges such as the lack of ground-truth labels.

**Neural combinatorial optimization (NCO)**   While both our framework and NCO address NP-hard problems, they differ fundamentally in motivation and goals. NCO focuses on efficiently finding

---

[1]Many classical primal-dual algorithms achieve tight worst-case approximation bounds under the Unique Games Conjecture (e.g., Khot & Regev, 2008). While worst-case bounds cannot be exceeded, our empirical results demonstrate improved performance in a beyond-worst-case setting.

optimal or near-optimal solutions by leveraging neural networks to learn task-specific heuristics or end-to-end optimization (Dai et al., 2018; Li et al., 2018; Joshi et al., 2019; Karalias & Loukas, 2021; Wang & Li, 2023). In contrast, NAR emphasizes designing neural architectures that replicate the reasoning processes of classical algorithms, with a focus on generalizing algorithmic behavior across diverse problem sizes and structures. Nonetheless, our framework also contributes to NCO, addressing an underexplored aspect by proposing an algorithmically informed GNN that tackles challenges of data efficiency and generalization. We discuss this connection in detail in Appendix H.

# 3 PROBLEM STATEMENT

We focus on studying three NP-hard problems: Minimum Vertex Cover, Minimum Set Cover, and Minimum Hitting Set (Section 3.1). We then illustrate a general primal-dual approximation algorithm using the minimum hitting set problem (Section 3.2).

## 3.1 NP-HARD PROBLEMS

**Definition 1** (Minimum Vertex Cover). *Let $G = (V, E)$ be a graph where $V$ are vertices and $E$ are edges, and each vertex $v \in V$ has a non-negative weight $w_v \in \mathbb{R}^+$. A vertex cover for $G$ is a subset $C \subseteq V$ of the vertices such that for each edge $(v, u) \in E$, either $v \in C$, $u \in C$, or both. The objective is to minimize the total vertex weight $\sum_{v \in C} w_v$.*

**Definition 2** (Minimum Set Cover). *Given a ground set $\mathcal{U}$ and a family of sets $\mathcal{C} \subseteq 2^{\mathcal{U}}$ with non-negative weights $w_S \in \mathbb{R}^+$ for all sets $S \in \mathcal{C}$, a set cover is a subfamily $\mathcal{C}' \subseteq \mathcal{C}$ such that $\cup_{S \in \mathcal{C}'} S = \cup_{S \in \mathcal{C}} S$. The objective is to minimize the total weight $\sum_{S \in \mathcal{C}'} w_S$.*

**Definition 3** (Minimum Hitting Set). *Given a ground set $E$ of elements $e$ with non-negative weights $w_e \in \mathbb{R}^+$ and a collection $\mathcal{T}$ of subsets $T \subseteq E$, a hitting set is a subset $A \subseteq E$ such that $A \cap T \neq \emptyset$ for every $T \in \mathcal{T}$. The objective is to minimize the total weight $\sum_{e \in A} w_e$.*

The minimum vertex cover problem is a foundational NP-hard problem with wide-reaching applications. As we will describe formally in Section 3.2, its dual problem is the well-studied *maximum edge-packing problem*. This primal-dual pair inspires a famous 2-approximation algorithm proposed by Hochbaum (1982) and many follow-up works. The minimum set cover problem is a generalization of vertex cover to hypergraphs, making it critical for understanding optimization over more complex structures. Lastly, the hitting set is equivalent to the minimum set cover problem, but its formulation more naturally extends to a wide range of combinatorial optimization problems, including vertex cover, Steiner tree, feedback vertex set, and many more (Goemans & Williamson, 1996).

## 3.2 A GENERAL PRIMAL-DUAL APPROXIMATION ALGORITHM

We now illustrate a general primal-dual approximation algorithm using the hitting set problem. Due to the problem's generality, the algorithm applies to any problem representable by the hitting set and can be generalized to other problems as well (Williamson & Shmoys, 2011). Many combinatorial optimization problems are naturally expressed as *integer programs (IPs)*, where variables are restricted to integer values. The IP formulation of the minimum hitting set (MHS) is shown in Figure 1(a).

A common approach to designing approximation algorithms is to use the IP's *linear programming (LP)* relaxation. This technique relaxes the integer constraints and allows variables to take continuous values, making the problem more tractable. Many of the best-known approximation algorithms utilize LP relaxations to derive solutions that can be efficiently rounded to obtain integer solutions. Furthermore, every LP formulation has a dual version. The primal-dual pair for MHS is shown in Figure 1(b) and Figure 1(c). Figure 1(b) is the *primal* and Figure 1(c) is the *dual*. More generally, the dual of an LP $\min_{\boldsymbol{x} \geq \boldsymbol{0}} \{\boldsymbol{c}^\top \boldsymbol{x} : A\boldsymbol{x} \geq \boldsymbol{b}\}$ is defined as $\max_{\boldsymbol{y} \geq \boldsymbol{0}} \{\boldsymbol{y}^\top \boldsymbol{b} : A^T \boldsymbol{y} \leq \boldsymbol{c}\}$. It is often useful to incorporate the LP relaxation's dual formulation to design better approximation algorithms. The *weak duality principal* states that any feasible solution to the primal problem has a larger objective value than any feasible solution to the dual problem. Moreover, the *strong duality principle* states that if an LP has an optimal solution, then the optimal value of the primal problem is equal to the

$$\text{Minimize} \quad \sum_{e \in E} w_e x_e$$
$$\text{subject to} \quad \sum_{e \in T} x_e \geq 1, \quad \forall T \in \mathcal{T}$$
$$x_e \in \{0,1\}, \quad \forall e \in E$$

$$\text{Minimize} \quad \sum_{e \in E} w_e x_e$$
$$\text{subject to} \quad \sum_{e \in T} x_e \geq 1, \quad \forall T \in \mathcal{T}$$
$$x_e \geq 0, \quad \forall e \in E$$

$$\text{Maximize} \quad \sum_{T \in \mathcal{T}} y_T$$
$$\text{subject to} \quad \sum_{T:e \in T} y_T \leq w_e, \quad e \in E$$
$$y_T \geq 0, \quad \forall T \in \mathcal{T}$$

(a) Minimum Hitting Set (MHS)  (b) LP relaxation of MHS  (c) Dual of LP relaxation of MHS

Figure 1: Let $x_e \in \{0,1\}$ for each element $e \in E$ be the variables, where $x_e = 1$ represents that element $e$ is included in the hitting set $A$, the IP formulation of MHS is shown in (a). Let $x_e \in \mathbb{R}^+$ be the primal variables, the LP relaxation of MHS is shown in (b). Let $y_T \in \mathbb{R}^+$ for each set $T \in \mathcal{T}$ be the dual variables, the dual problem of the LP relaxation of MHS is shown in (c).

optimal value of its dual. Based on these principles, the primal-dual framework iteratively updates both the primal and dual solutions, closing their gap and ensuring they improve in tandem.

---

**Algorithm 1** General primal-dual approximation algorithm (with uniform increase)

---

**Input:** Ground set $E$ with weights $\boldsymbol{w}$, family of subsets $\mathcal{T} \subseteq 2^E$
1: $A \leftarrow \emptyset$; for all $e \in E, r_e \leftarrow w_e$
2: **while** $\exists T : A \cap T = \emptyset$ **do**
3: $\quad \mathcal{V} \leftarrow \{T : A \cap T = \emptyset\}$
4: $\quad$ **repeat**
5: $\quad\quad$ **for** $T \in \mathcal{V}$ **do** $\delta_T \leftarrow \min_{e \in T} \left\{ \frac{r_e}{|\{T':e \in T'\}|} \right\}$ $\qquad$ Uniform increase:
6: $\quad\quad$ **for** $e \in E \setminus A$ **do** $r_e \leftarrow r_e - \sum_{T:e \in T} \delta_T$ $\qquad$ (6.1): $\Delta \leftarrow \min_{T \in \mathcal{V}} \delta_T$
7: $\quad$ **until** $\exists e \notin A : r_e = 0$ $\qquad\qquad\qquad\qquad\qquad$ (6.2): **for** $e \in E \setminus A$ **do**
8: $\quad A \leftarrow A \cup \{e : r_e = 0\}$ $\qquad\qquad\qquad\qquad\qquad\qquad r_e \leftarrow r_e - |\{T : e \in T\}|\Delta$
**Output:** $A$

---

Based on the primal-dual framework, an $\alpha$-approximation algorithm (Bar-Yehuda & Even, 1981; Hochbaum, 1982; Goemans & Williamson, 1996; Khuller et al., 1994) for the general hitting set problem was developed, where $\alpha$ is the maximal cardinality of the subsets. The pseudocode of the algorithm is shown in Algorithm 1. Given a hitting set problem $(\mathcal{T}, E, w)$, the algorithm progresses over a series of rounds. At each round, the algorithm increases some of the dual variables $y_T$ until a constraint $\sum_{T:e \in T} y_T \leq w_e$ becomes an equality, at which point the element $e$ is added to the hitting set $A$. Although the algorithm does not explicitly define the dual variables $y_T$, it can be interpreted as gradually increasing the dual variables by an amount $\delta_T$ in each round, as shown in Line 5. This is implemented by defining a residual weight $r_e = w_e - \sum_{T:e \in T} y_T$, which is defined in terms of the step sizes $\delta_T$ in Line 6. Once $r_e = 0$ for some $e \notin A$ (i.e. the constraint becomes tight), $e$ is added to the hitting set $A$ (Lines 7 and 8). This process is repeated until $A$ is a valid hitting set (Line 2).

**A general framework** This algorithm can be reformulated to recover many classical (exact or approximation) algorithms for problems that are special cases of the hitting set problem (Goemans & Williamson, 1996). For example, vertex cover can be seen as a hitting set problem, where each element $e \in E$ corresponds to each vertex $v \in V$, and each subset $T \in \mathcal{T}$ corresponds to an edge that connects two vertices. This allows a direct adaptation of Algorithm 1 to solve the vertex cover problem. Moreover, Khuller et al. (1994) propose a sublinear-time vertex cover approximation algorithm which is a simple generalization of Algorithm 1. They relax the dual constraint using a parameter $\epsilon > 0$, such that a vertex $e$ is included in the cover if $r_e \leq \epsilon w_e$, instead of $r_e = 0$. This results in a $2/(1-\epsilon)$-approximation algorithm with a time complexity of $O(\ln^2 |\mathcal{T}| \ln \frac{1}{\epsilon})$. Since set cover extends vertex cover to hypergraphs, this algorithm can be adapted into an $r/(1-\epsilon)$-approximation algorithm for set cover, where $r$ is the maximal cardinality of the sets. A more detailed explanation of vertex cover and set cover, along with their algorithms, is provided in Appendix A.

**Uniform increase of dual variables** For some problems, it is beneficial to simultaneously increase all dual variables $\delta_T$ at the same rate (Agrawal et al., 1995; Goemans & Williamson, 1995). An

example is the minimum spanning tree problem, which is a special case of the hitting set problem. Kruskal's algorithm (Kruskal, 1956) for the problem greedily selects the minimum-cost edge that connects two distinct components. This corresponds to increasing the dual variables for all connected components simultaneously, until there is an edge whose dual constraint becomes tight. The uniform increase rule provides a more balanced approach to attend to all dual variables. To incorporate this uniform increase rule, Line 6 of the algorithm is modified as highlighted in red. The optional addition of the uniform increase rule allows the framework to adapt to a broader range of algorithms.

## 4 PRIMAL-DUAL GRAPH NEURAL NETWORKS (PDGNNS)

We now present our framework of using a GNN to simulate the general primal-dual approximation algorithm, representing the primal-dual variables as two sides in a bipartite graph (Section 4.1). We also show how the uniform increase rule can be incorporated with a virtual node that connects to all dual nodes in the bipartite graph (Section 4.2). Furthermore, we explain how we use optimal solutions from integer programming solvers as additional training signals (Section 4.3), and later show how it allows the PDGNN to surpass the performance of the approximation algorithm via experiments.

### 4.1 ARCHITECTURE

We adopt the encoder-processor-decoder framework (Hamrick et al., 2018) from the neural algorithmic reasoning blueprint (Veličković & Blundell, 2021) to simulate Algorithm 1 with hitting set as an example. In this framework, the *processor* is typically a message-passing GNN (Gilmer et al., 2017) that operates within a latent space. The *encoder* transforms the input data into this latent space, while the *decoder* reconstructs the final output from it.

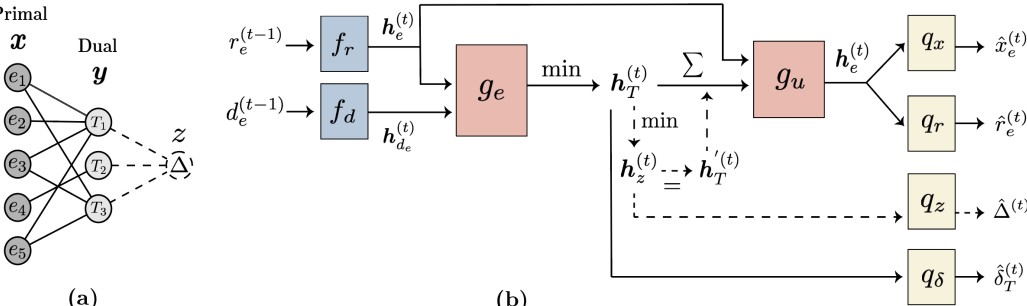

(a)                                (b)

Figure 2: (a) Bipartite graph construction. (b) The architecture of PDGNN with the encoder, processor, and decoder colored distinctively. $\Delta$ is only used when the uniform increase rule is applied.

**Bipartite graph construction**    Given a hitting set problem $(\mathcal{T}, E, w)$, we represent it as a bipartite graph with elements $e \in E$ (primal) on the left-hand side (LHS) and sets $T \in \mathcal{T}$ (dual) on the right-hand side (RHS), as illustrated in Figure 2. An edge is created between an element $e$ and a set $T$ if $e \in T$. Let $\mathcal{N}(e)$ denote the set of neighbors of node $e$. As outlined in Algorithm 1, the algorithm incrementally adds elements $e$ to the hitting set $A$. When an element is added, we remove node $e$ by masking it along with its neighboring sets $T \in \mathcal{N}(e)$, which are now hitted. Consequently, the violation set $\mathcal{V} = \{T \mid A \cap T = \emptyset\}$ consists of the remaining sets $T$ still in the graph. Once $A$ becomes a valid hitting set, the violation set $\mathcal{V}$ is empty. Next, at each timestep $t$, let $r_e^{(t)}$ denote the residual weight of element $e$ and $d_e^{(t)}$ denote its current node degree. The initial residual weight $r_e^{(0)}$ is defined as its cost $w_e$, while the initial degree $d_e^{(0)}$ is given by $|\{T : e \in T\}|$.

**Encoder**    The architecture includes two MLP encoders, $f_r$ and $f_d$, which encode the residual node weight $r_e^{(t)}$ and node degree $d_e^{(t)}$, respectively, for each element $e \in E$. These encoders transform all features into a high-dimensional latent space for the processor:

$$\boldsymbol{h}_e^{(t)} = f_r(r_e^{(t-1)}), \qquad\qquad\qquad \boldsymbol{h}_{d_e}^{(t)} = f_d(d_e^{(t-1)}).$$

**Processor** The processor is a message-passing GNN applied to the bipartite graph. A general message-passing framework (Gilmer et al., 2017) comprises a message function $\psi_\theta$ and an update function $\phi_\theta$. The node feature $\boldsymbol{h}_v^{(t)}$ of node $v$ is transformed via $\boldsymbol{h}_v^{(t)} = \phi_\theta(\boldsymbol{h}_v^{(t)}, \bigoplus_{u \in \mathcal{N}(v)} \psi_\theta(\boldsymbol{h}_u^{(t)}))$, where $\psi_\theta$ and $\phi_\theta$ are usually shallow MLPs, and $\bigoplus$ is a permutation invariant function, such as sum or max. We now demonstrate how this message-passing framework is applied to the bipartite graph to simulate the primal-dual approximation algorithm.

Step (1): This step corresponds to Line 5 of Algorithm 1, where increment $\delta_T^{(t)}$ for each set $T \in \mathcal{V}$ is computed. Let $\boldsymbol{h}_T^{(t)}$ be the hidden representation of $\delta_T^{(t)}$. We aggregate messages from its connected elements $e \in \mathcal{N}(T)$ using a message function $g_e$ with a min aggregation operation:

$$\boldsymbol{h}_T^{(t)} = \min_{e \in \mathcal{N}(T)} g_e(\boldsymbol{h}_e^{(t)}, \boldsymbol{h}_{d_e}^{(t)}).$$

Step (2): This step corresponds to Line 6 of Algorithm 1, where residual weight $r_e^{(t)}$ for each element $e \in E \setminus A$ is computed. Therefore, the dual variable update $\boldsymbol{h}_T^{(t)}$ is passed back to its connected elements $e$ using a sum aggregation and an update function $g_u$:

$$\boldsymbol{h}_e^{(t)} = g_u \left( \boldsymbol{h}_e^{(t)}, \sum_{T \in \mathcal{N}(e)} \boldsymbol{h}_T^{(t)} \right).$$

**Decoder** At each timestep $t$, Algorithm 1 computes three types of intermediate quantities: (1) whether to include an element $e$ in the hitting set, represented by $x_e^{(t)} \in \{0, 1\}$, (2) the residual weights of an element $r_e^{(t)}$, and (3) the increment to the dual variable $\delta_T^{(t)}$. We utilize separate MLP decoders, $q_x$, $q_r$, and $q_\delta$, to compute each of these quantities:

$$\hat{x}_e^{(t)} = q_x(\boldsymbol{h}_e^{(t)}), \qquad\qquad \hat{r}_e^{(t)} = q_r(\boldsymbol{h}_e^{(t)}), \qquad\qquad \hat{\delta}_T^{(t)} = q_\delta(\boldsymbol{h}_T^{(t)}).$$

**Training** Given the recurrent nature of our architecture, we apply *noisy teacher forcing* (Veličković et al., 2022) with a probability of 0.5 to determine whether to use *hints*—ground-truth values for intermediate quantities above—as inputs for the next timestep. Otherwise, the model's prediction from the previous timestep is passed on as the inputs. This approach allows the model to follow its recurrent flow while reducing the risk of error propagation. For the node masks, however, teacher forcing is always applied since intermediate targets $x_e^{(t)}$, $r_e^{(t)}$, and $\delta_T^{(t)}$ are only available for nodes that have not yet been removed by Algorithm 1. The entire encoder-processor-decoder cycle is repeated for a maximum of $|E|$ timesteps or terminates early when the solution becomes a valid hitting set. The loss function is defined as $\mathcal{L}_{\text{algo}}^{(t)} = \mathcal{L}_{\text{BCE}}(\hat{x}_e^{(t)}, x_e^{(t)}) + \mathcal{L}_{\text{MSE}}(\hat{r}_e^{(t)}, r_e^{(t)}) + \mathcal{L}_{\text{MSE}}(\hat{\delta}_T^{(t)}, \delta_T^{(t)})$ and averaged across timesteps. During test time, if the model output does not produce a valid hitting set, a cleanup stage is employed, where we greedily add the element $e$ with the highest $r_e/d_e$ value to the solution. We will discuss the frequency of cleanup in the experiments section.

## 4.2 UNIFORM INCREASE OF DUAL VARIABLES

The uniform increase rule requires global communication among all dual variables. To achieve this, we introduce a virtual node $z$ that connects to every set $T \in \mathcal{T}$, as shown in Figure 2. Below, we describe how Step (2) in the processor is adjusted to accommodate this modification.

- Step (2.1): The virtual node aggregates all messages from the dual variables $\boldsymbol{h}_T^{(t)}$ via a min aggregation, corresponding to Line 6.1 in Algorithm 1 via $\boldsymbol{h}_z^{(t)} = \min_{T \in \mathcal{T}} \boldsymbol{h}_T^{(t)}$.

- Step (2.2): The global information is passed back to dual variables with temporary $\boldsymbol{h}_T'^{(t)} = \boldsymbol{h}_z^{(t)}$, and then to the primal variables $\boldsymbol{h}_e^{(t)}$ with an update function $g_u$ and a sum aggregation. This corresponds to Line 6.2 in Algorithm 1 via $\boldsymbol{h}_e^{(t)} = g_u(\boldsymbol{h}_e^{(t)}, \sum_{T \in \mathcal{N}(e)} \boldsymbol{h}_T'^{(t)})$.

The intermediate quantity $\Delta^{(t)}$ is also given by Algorithm 1. We use an additional decoder $q_\Delta$ to compute the prediction $\hat{\Delta}^{(t)} = q_\Delta(\boldsymbol{h}_z^{(t)})$ and add $\mathcal{L}_{\text{MSE}}(\hat{\Delta}^{(t)}, \Delta^{(t)})$ to the total loss $\mathcal{L}_{\text{algo}}^{(t)}$.

**Theoretical justification**    Our model architecture is designed to closely align with the primal-dual approximation algorithm. The following theorem shows that PDGNN can replicate the behavior of Algorithm 1. A detailed proof is provided in in Appendix B.

**Theorem 1.** *Given a hitting set problem* $(\mathcal{T}, E, w)$, *let* $\mathcal{A}(\mathcal{T}, E, w)$ *be the solution produced by Algorithm 1, which terminates after* $K$ *timesteps. There exists a parameter configuration* $\Theta$ *for a PDGNN model* $\mathcal{M}_\Theta$ *such that, at timestep* $K$, *the model output satisfies* $\mathcal{M}_\Theta^{(K)}(\mathcal{T}, E, w) = \mathcal{A}(\mathcal{T}, E, w)$. *Furthermore, let* $(\boldsymbol{x}^{(t)}, \boldsymbol{r}^{(t)}, \boldsymbol{\delta}^{(t)}, \Delta^{(t)})$ *be the intermediate quantities computed by Algorithm 1 at each timestep* $t$. *Then, the PDGNN model satisfies* $\mathcal{M}_\Theta^{(t)}(\mathcal{T}, E, w) = (\boldsymbol{x}^{(t)}, \boldsymbol{r}^{(t)}, \boldsymbol{\delta}^{(t)}, \Delta^{(t)})$.

### 4.3    USE OF OPTIMAL SOLUTIONS FROM SOLVERS

The primal-dual algorithm is an approximation algorithm, while optimal solutions can be computed on small instances using IP solvers. We use the default IP solver in `scipy`, which is based on HiGHS (Schwendinger & Schumacher, 2023; Huangfu & Hall, 2018). These optimal solutions are used as additional training signals to guide the model toward better outcomes. However, unlike the primal-dual algorithm, which provides intermediate steps, IP solvers only produce the final optimal solution. Therefore, the corresponding loss is defined as $\mathcal{L}_{\text{optm}} = \mathcal{L}_{\text{BCE}}(\hat{x}_e^K, x_e^{\text{optm}})$, where $K$ is the final timestep. The overall loss is then the sum of the intermediate losses from the primal-dual algorithm and the optimal solution loss, given by $\mathcal{L} = \frac{1}{K} \sum_{t=1}^{K} \mathcal{L}^{(t)}\text{algo} + \mathcal{L}_{\text{optm}}$. The motivation stems from the fact that IP solvers are computationally expensive, especially for larger problem instances. By training PDGNN using optimal solutions from IP solvers on smaller problem instances—allowing it to exceed the performance of the approximation algorithm—we can leverage its generalization ability to create a cost-efficient, high-performance model for much larger instances.

## 5    EXPERIMENTS

### 5.1    SYNTHETIC NP-HARD PROBLEMS

**Dataset**    We evaluate PDGNN on three NP-hard problems as described in Section 3.1. The training dataset includes 1000 random graphs of size 16: Barabási-Albert graphs for vertex cover and random bipartite graphs for set cover and hitting set. Generalization is tested on 100 graphs of sizes 16, 32, and 64, with each experiment repeated across 10 seeds.

**Baselines**    PDGNN leverages both the intermediate steps of the algorithm and the optimal solutions provided by IP solvers. To demonstrate its effectiveness, we compare PDGNN against two variants: one without intermediate steps (No algo) and another without optimal solutions (No optm). Additionally, PDGNN's aggregation strategy is specifically tailored to align with the algorithm's structure (see Section 4.1). To validate this design, we test alternative aggregation methods, such as Mean and Max pooling. Lastly, since PDGNN has a recurrent structure, we compare it to a powerful end-to-end GNN trained on optimal solutions as a node classification task. For this, we use a 3-layer GIN model (Xu et al., 2019) with jumping knowledge (Xu et al., 2018).

Results are summarized in Table 1. The objective is to minimize the total weight of the elements in the solution. The ratio $w_{\text{pred}}/w_{\text{appr}}$ represents the predicted solution weight relative to that of the primal-dual approximation, where a ratio below 1.0 indicates superior model performance. Similarly, $w_{\text{pred}}/w_{\text{optm}}$ compares the predicted weights to the optimal solution, with a ratio closer to 1.0 being ideal. Finally, the percentage of sets that remain uncovered or unhit by the predicted solutions (Unhit) reflects how often the cleanup stage is required. All weights are calculated after the cleanup stage to ensure the validity of the model's predictions.

**PDGNN is the most effective**    We find that combining losses from both intermediate steps of the approximation algorithm and optimal solutions of IP solvers enables PDGNN to achieve the best performance, yielding the lowest ratios across all test cases. Additionally, PDGNN's performance remains stable across different graph sizes, with a low percentage of uncovered sets, indicating strong generalization to larger graphs. Comparisons with other baselines show that incorporating supervision from optimal solutions improves the quality of final predictions, allowing PDGNN to outperform the primal-dual algorithm it was designed to simulate. In contrast, training without intermediate

Table 1: Model performance across different graph sizes for Minimum Vertex Cover (MVC), Minimum Set Cover (MSC), and Minimum Hitting Set (MHS).

| | Method | 16 nodes | | 32 nodes | | 64 nodes | | Unhit |
|---|---|---|---|---|---|---|---|---|
| | | $w_{pred} / w_{algo}$ | $w_{pred} / w_{optm}$ | $w_{pred} / w_{algo}$ | $w_{pred} / w_{optm}$ | $w_{pred} / w_{algo}$ | $w_{pred} / w_{optm}$ | % |
| MVC | GIN | $0.988 \pm 0.009$ | $1.154 \pm 0.011$ | $1.068 \pm 0.068$ | $1.218 \pm 0.077$ | $1.124 \pm 0.106$ | $1.279 \pm 0.120$ | 2.66 |
| | No algo | $0.968 \pm 0.038$ | $1.129 \pm 0.044$ | $1.074 \pm 0.039$ | $1.224 \pm 0.044$ | $1.106 \pm 0.031$ | $1.259 \pm 0.035$ | 0.30 |
| | No optm | $0.994 \pm 0.004$ | $1.160 \pm 0.005$ | $0.997 \pm 0.005$ | $1.137 \pm 0.006$ | $1.000 \pm 0.008$ | $1.138 \pm 0.009$ | 1.71 |
| | Mean | $1.035 \pm 0.026$ | $1.208 \pm 0.030$ | $1.081 \pm 0.032$ | $1.233 \pm 0.037$ | $1.106 \pm 0.044$ | $1.259 \pm 0.051$ | 2.54 |
| | Max | $0.972 \pm 0.024$ | $1.134 \pm 0.028$ | $1.001 \pm 0.022$ | $1.141 \pm 0.025$ | $1.000 \pm 0.021$ | $1.138 \pm 0.023$ | 1.87 |
| | PDGNN | $\mathbf{0.945} \pm 0.004$ | $\mathbf{1.103} \pm 0.004$ | $\mathbf{0.958} \pm 0.004$ | $\mathbf{1.092} \pm 0.005$ | $\mathbf{0.967} \pm 0.005$ | $\mathbf{1.100} \pm 0.005$ | 0.79 |
| MSC | No algo | $1.077 \pm 0.016$ | $1.159 \pm 0.017$ | $1.142 \pm 0.057$ | $1.265 \pm 0.063$ | $3.544 \pm 1.305$ | $4.197 \pm 1.544$ | 3.98 |
| | No optm | $1.007 \pm 0.004$ | $1.086 \pm 0.005$ | $1.018 \pm 0.004$ | $1.127 \pm 0.005$ | $1.025 \pm 0.006$ | $1.214 \pm 0.007$ | 0.00 |
| | PDGNN | $\mathbf{0.997} \pm 0.003$ | $\mathbf{1.075} \pm 0.004$ | $\mathbf{0.996} \pm 0.007$ | $\mathbf{1.103} \pm 0.008$ | $\mathbf{1.003} \pm 0.015$ | $\mathbf{1.188} \pm 0.018$ | 0.14 |
| MHS | No algo | $1.075 \pm 0.013$ | $1.157 \pm 0.014$ | $1.120 \pm 0.042$ | $1.265 \pm 0.047$ | $3.704 \pm 0.796$ | $4.459 \pm 0.957$ | 5.62 |
| | No optm | $0.998 \pm 0.001$ | $1.074 \pm 0.001$ | $0.997 \pm 0.002$ | $1.126 \pm 0.002$ | $0.991 \pm 0.003$ | $1.193 \pm 0.004$ | 0.00 |
| | PDGNN | $\mathbf{0.995} \pm 0.002$ | $\mathbf{1.071} \pm 0.002$ | $\mathbf{0.987} \pm 0.002$ | $\mathbf{1.114} \pm 0.002$ | $\mathbf{0.982} \pm 0.005$ | $\mathbf{1.182} \pm 0.007$ | 0.00 |

steps leads to a significant drop in generalization on larger graphs, highlighting that the primal-dual algorithm provides critical reasoning capabilities beyond merely learning optimal solution patterns.

**A general NAR framework** Our framework is effective across all three NP-hard problems, even when scaling to larger graphs. For the vertex cover problem, we demonstrate its applicability to graph-based problems. In the set cover problem, we extend the framework to hypergraphs using a bipartite structure, highlighting its flexibility beyond traditional graph settings. Additionally, the uniform increase rule proves effective for the hitting set problem, which serves as a general framework for a broad range of optimization tasks. Furthermore, these results suggest that our approach may achieve even better performance when trained on more challenging instances where the optimal solutions significantly outperform those of approximation algorithms.

## 5.2 GENERALIZATION TO OOD GRAPH FAMILY

**Dataset** We evaluate the model's generalization on out-of-distribution (OOD) graph families using the vertex cover problem. The training set consists of 1000 Barabási-Albert (B-A) graphs with 16 nodes, while each test set contains 100 graphs from different families: Erdős–Rényi (E-R), Star, Lobster, and 3-connected planar (3-Con) graphs. These graph types pose unique challenges for the vertex cover problem due to their distinct structural properties. Results are averaged across 10 seeds.

Table 2: Model performance on OOD graph types for the minimum vertex cover problem.

| Type | 16 nodes | | 32 nodes | | 64 nodes | | Unhit |
|---|---|---|---|---|---|---|---|
| | $w_{pred} / w_{algo}$ | $w_{pred} / w_{optm}$ | $w_{pred} / w_{algo}$ | $w_{pred} / w_{optm}$ | $w_{pred} / w_{algo}$ | $w_{pred} / w_{optm}$ | % |
| E-R | $0.949 \pm 0.005$ | $1.084 \pm 0.006$ | $0.961 \pm 0.004$ | $1.098 \pm 0.005$ | $0.961 \pm 0.005$ | $1.088 \pm 0.006$ | 7.92 |
| Star | $0.948 \pm 0.004$ | $1.078 \pm 0.004$ | $0.958 \pm 0.004$ | $1.096 \pm 0.005$ | $0.962 \pm 0.007$ | $1.092 \pm 0.008$ | 7.16 |
| Lobster | $0.939 \pm 0.004$ | $1.090 \pm 0.004$ | $0.957 \pm 0.003$ | $1.093 \pm 0.004$ | $0.962 \pm 0.007$ | $1.095 \pm 0.007$ | 8.78 |
| 3-Con | $0.945 \pm 0.002$ | $1.087 \pm 0.002$ | $0.959 \pm 0.005$ | $1.085 \pm 0.005$ | $0.955 \pm 0.005$ | $1.085 \pm 0.006$ | 7.76 |

Table 2 shows that PDGNN consistently outperforms the approximation algorithm across all graph families and sizes, demonstrating strong generalization despite being trained only on Barabási-Albert graphs of size 16. Notably, these graph types have vastly different optimal vertex cover sizes: Erdős–Rényi graphs require an average of 80% of nodes, while Star graphs need only 15% (see in Table 5). These results further highlight the robustness of our model.

## 5.3 COMMERCIAL OPTIMIZATION SOLVERS

One practical use case for PDGNN is to warm start large-scale commercial solvers, such as Gurobi (Gurobi Optimization, LLC, 2024), by initializing variables with its predictions. The motivation is that providing a starting point closer to the optimal solution can lead to faster solving times

and improved efficiency. We evaluate the vertex cover problem by comparing solutions from the primal-dual algorithm and our model as warm starts for Gurobi against its default initialization. The model, trained on 1000 B-A graphs of size 16, produces solutions on random B-A graphs of sizes 500, 600, and 750, with 100 graphs per size. Results are averaged over 5 seeds. We use the default parameters for Gurobi, setting the thread count to 1 and imposing a time limit of 1 hour.

**Metrics** We report the number of wins in solving time and the total number of cases solved (Wins / Solved), the 1-shifted geometric mean of solving times (Total time) as a standard metric (Gasse et al., 2019a) (which includes cases when the time limit is hit but the optimal solution is not found), the arithmetic mean of solving times only when solved to optimality (Opt. time) (Huang et al., 2024) (which only takes the average of those cases where optimal solutions are found), and the number of nodes explored in the branch-and-bound tree (B&B nodes). Additionally, we measure the average computation time to generate the warm-start solutions using the primal-dual algorithm and the model per graph (Comp. time). All times are reported in seconds.

Table 3: Performance of using solutions from a PDGNN trained on 16-node graphs and the primal-dual approximation algorithm to warm start Gurobi.

| #Nodes | Method | Wins / Solved | Total time (geometric) | Opt. time (arithmetic) | B&B nodes | Comp. time |
|---|---|---|---|---|---|---|
| 500 | None | 15.0 / 100.0 | $4.33 \pm 0.06$ | $78.97 \pm 2.24$ | $3142.27 \pm 290.00$ | - |
| | Algorithm | 33.4 / 100.0 | $4.25 \pm 0.03$ | $76.69 \pm 1.54$ | $3121.60 \pm 85.61$ | 0.20 |
| | PDGNN | **51.6 / 100.0** | **4.15** $\pm 0.02$ | **72.61** $\pm 2.13$ | **3047.86** $\pm 185.67$ | 0.02 |
| 600 | None | 20.4 / 98.0 | $11.63 \pm 0.05$ | $230.84 \pm 19.59$ | $7793.87 \pm 301.70$ | - |
| | Algorithm | 31.4 / 98.0 | $11.57 \pm 0.13$ | $227.83 \pm 19.32$ | $7801.66 \pm 288.23$ | 0.30 |
| | PDGNN | **46.4 / 98.0** | **11.34** $\pm 0.09$ | **209.52** $\pm 14.15$ | **7594.29** $\pm 198.16$ | 0.03 |
| 750 | None | 13.6 / 80.8 | $15.87 \pm 0.07$ | $299.01 \pm 23.53$ | $9844.53 \pm 182.05$ | - |
| | Algorithm | 25.6 / 80.8 | $15.74 \pm 0.12$ | $300.12 \pm 22.89$ | $9838.71 \pm 151.04$ | 0.39 |
| | PDGNN | **42.4 / 80.8** | **15.69** $\pm 0.11$ | **292.92** $\pm 18.61$ | **9749.62** $\pm 136.19$ | 0.03 |

Table 3 shows that PDGNN outperforms both the default initialization and the approximation algorithm in all cases, achieving the fastest mean solving time and exploring fewer nodes in the search tree. The improvement from using the model over the algorithm is greater than that of the algorithm over no warm start. Additionally, the model's inference time is nearly 10 times faster than the approximation algorithm's computation time. This demonstrates a practical use case: by simulating an approximation algorithm and leveraging optimal solutions on small instances, the model generates high-quality solutions for larger problems, reducing solving time and B&B tree size for large-scale commercial solvers, such as Gurobi.

## 5.4 REAL-WORLD DATASETS

We present another use case for our model using real-world datasets. A key limitation of traditional algorithms designed for specific problems is that they cannot be applied directly to real-world data without preprocessing. For example, raw node features must typically undergo feature engineering to estimate vertex weights, reducing the problem to a minimum-weight vertex cover before using the algorithm. In contrast, PDGNN can bypass this step by incorporating a new feature encoder and learning to estimate vertex weights directly from raw data. We demonstrate this advantage through experiments on three real-world datasets: Airports (Brazil, Europe, USA) (Ribeiro et al., 2017; Jin et al., 2019), WikipediaNetwork (Chameleon, Squirrel) (Rozemberczki et al., 2019; Pei et al., 2020), and PPI (Zitnik & Leskovec, 2017). These datasets are chosen because their classification targets relate to the notion of node influences, where vertex cover may provide valuable insights.

**Architecture and baselines** We evaluate three base models: GCN (Kipf & Welling, 2017), GAT (Veličković et al., 2018), and GraphSAGE (Hamilton et al., 2017), applying them to the node classification datasets using standard procedures. Recall we follow the encoder-processor-decoder framework from NAR. We then use the pretrained PDGNN from Section 5.1 on B-A graphs of size 16, keeping only the processor and degree encoder. We trains a new encoder learns to map node features

into the shared latent space of the processor, enabling it to replicate vertex cover problem-solving behavior. The pretrained components are frozen, and a single message-passing step is performed, similar to Numeroso et al. (2023). PDGNN's output embeddings are then concatenated with the base model's output embeddings before passing through a linear layer for final classification. We compare PDGNN against two baselines: Node2Vec (Grover & Leskovec, 2016) and a degree encoder to demonstrate that PDGNN captures more complex information beyond node degrees. Node2Vec is applied only to transductive datasets. All experiments are repeated 10 times.

Table 4: Performance of embeddings generated using PDGNN, Node2Vec (N2V), and degree encoder (Degree) with three base models. All results are reported in percentages.

| | Airports (Accuracy) | | | WikipediaNetwork (Accuracy) | | PPI (Micro-F1) |
|---|---|---|---|---|---|---|
| | Brazil | Europe | USA | Chameleon | Squirrel | PPI |
| GCN | 58.89 $\pm_{17.72}$ | 63.87 $\pm_{10.74}$ | 77.48 $\pm_{2.28}$ | 62.57 $\pm_{2.05}$ | 52.51 $\pm_{1.26}$ | 58.55 $\pm_{0.36}$ |
| GCN+Degree | 71.48 $\pm_{15.53}$ | 70.37 $\pm_{4.16}$ | 79.25 $\pm_{2.16}$ | 63.22 $\pm_{1.75}$ | 53.78 $\pm_{1.59}$ | 59.21 $\pm_{0.71}$ |
| GCN+N2V | 73.33 $\pm_{5.44}$ | 74.75 $\pm_{5.12}$ | 79.54 $\pm_{2.19}$ | 64.04 $\pm_{1.87}$ | **54.97** $\pm_{1.52}$ | - |
| GCN+PDGNN | **81.11** $\pm_{9.46}$ | **76.88** $\pm_{5.10}$ | **82.82** $\pm_{3.06}$ | **64.58** $\pm_{2.17}$ | 53.53 $\pm_{1.32}$ | **60.93** $\pm_{0.92}$ |
| GAT | 60.13 $\pm_{15.68}$ | 65.85 $\pm_{9.68}$ | 80.59 $\pm_{2.43}$ | 65.18 $\pm_{1.95}$ | 53.48 $\pm_{1.37}$ | 56.73 $\pm_{0.87}$ |
| GAT+Degree | 66.30 $\pm_{12.27}$ | 77.75 $\pm_{4.10}$ | 82.82 $\pm_{2.08}$ | 65.92 $\pm_{2.18}$ | 55.03 $\pm_{2.02}$ | 57.51 $\pm_{1.31}$ |
| GAT+N2V | 75.56 $\pm_{9.82}$ | 75.87 $\pm_{5.03}$ | 82.65 $\pm_{1.58}$ | 66.29 $\pm_{2.09}$ | 55.06 $\pm_{1.58}$ | - |
| GAT+PDGNN | **84.44** $\pm_{8.89}$ | **81.25** $\pm_{4.51}$ | **85.13** $\pm_{1.79}$ | **67.13** $\pm_{2.03}$ | **55.45** $\pm_{1.67}$ | **60.57** $\pm_{0.67}$ |
| SAGE | 44.82 $\pm_{17.40}$ | 61.82 $\pm_{10.38}$ | 78.07 $\pm_{3.39}$ | 60.42 $\pm_{1.85}$ | 41.20 $\pm_{1.52}$ | 61.87 $\pm_{0.46}$ |
| SAGE+Degree | 74.44 $\pm_{12.74}$ | 75.50 $\pm_{9.19}$ | 81.53 $\pm_{2.18}$ | 60.59 $\pm_{2.18}$ | 42.41 $\pm_{1.99}$ | 61.67 $\pm_{0.84}$ |
| SAGE+N2V | 80.00 $\pm_{6.46}$ | 76.12 $\pm_{3.75}$ | 83.15 $\pm_{3.20}$ | 60.09 $\pm_{2.25}$ | **42.88** $\pm_{1.59}$ | - |
| SAGE+PDGNN | **85.56** $\pm_{7.49}$ | **80.75** $\pm_{2.38}$ | **83.28** $\pm_{2.04}$ | **61.27** $\pm_{1.63}$ | 42.79 $\pm_{1.66}$ | **63.67** $\pm_{1.09}$ |

Table 4 shows that PDGNN achieves the highest improvements in most datasets. Note that Node2Vec has an additional advantage by directly training on the graphs to produce the embeddings. PDGNN's superior performance over the degree encoder also indicates that it captures more complex information by integrating both learned node weights from features and degree information. We observe that PDGNN is most effective in Airports and PPI, where node features are simpler. However, WikipediaNetwork datasets show less benefit from our model, which may be caused by the high dimensionality of node features (over 2000 dimensions). This significantly challenges the new encoder to learn to map complex features into the shared latent space with the pretrained processor, limiting its ability to generate meaningful vertex-cover embeddings. However, Numeroso et al. (2023) addressed similar issues by pretraining the new encoder on a dataset designed to reconstruct the target algorithm. Given the suitable datasets to have a similar pretraining setup, we believe PDGNN could potentially overcome this bottleneck and improve its performance.

## 6 CONCLUSIONS

We propose a novel and general NAR framework to learn algorithms for NP-hard problems. Our approach leverages both the intermediate solutions generated by primal-dual approximation algorithms and optimal solutions obtained from integer programming to train the model. While intermediate supervision from the algorithm provides a foundation for reasoning, incorporating optimal solutions enables the model to surpass the algorithm's performance. Empirical results demonstrate that our framework is effective and robust, showing strong generalization to larger graphs and OOD graph families. Additionally, we present two practical applications: warm starting commercial solvers for improved efficiency, and generating high-quality embeddings to enhance predictive performance on real-world datasets. Future work can expand our framework by incorporating additional techniques from Williamson & Shmoys (2011), which aid in designing more effective primal-dual approximation algorithms and further broaden its applicability.

## REPRODUCIBILITY STATEMENT

Our code can be downloaded at `https://anonymous.4open.science/r/pdgnn`. We also provide details of dataset generation in Appendix C, hardware and hyperparameter settings in Appendix D, and additional details of the architecture in Appendix E.

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

## A ADDITIONAL DETAILS OF VERTEX COVER AND SET COVER

### A.1 PRIMAL-DUAL PAIR: VERTEX COVER AND EDGE PACKING

Given a graph $G = (V, E)$, where $V$ are vertices and $E$ are edges, each vertex $v \in V$ has a non-negative weight $w : V \to \mathbb{R}^+$.

**Definition 4** (Minimum vertex cover). *A vertex-cover for $G$ is a subset $C \subseteq V$ of the vertices such that for each edge $(v, u) \in E$, either $v \in C$, $u \in C$, or both. The objective is to minimize the total vertex weight $\sum_{v \in C} w(v)$.*

**Definition 5** (Maximum edge packing). *An edge-packing is an assignment $p : E \to \mathbb{R}^+$ of non-negative weights to the edges $e \in E$, such that for any vertex $v \in V$, the total weight $\sum_{e:v \in e} p(e)$ assigned to the edges $e$ that are incident to $v$ is at most $w(v)$. The objective is to maximize the total edge weight $\sum_{e \in E} p(e)$.*

The edge packing problem is the dual of the LP relaxation of the vertex cover problem, which also has many practical implications, such as resource allocation. Because of this relationship, the primal-dual pair becomes key problems for studying approximation algorithms and the primal-dual framework. Let $x_v \in \{0, 1\}$ indicate whether each vertex $v \in V$ is in the cover, and $y_e \in \mathbb{R}^+$ be the non-negative weight assigned to each edge $e \in E$. The two problems can be formulated as:

$$
\begin{aligned}
\text{Min} \quad & \sum_{v \in V} w(v) x_v \\
\text{sub. to} \quad & x_u + x_v \geq 1, \ \forall e = (u, v) \in E \\
& x_v \in \{0, 1\}, \ \forall v \in V
\end{aligned}
\qquad
\begin{aligned}
\text{Max} \quad & \sum_{e \in E} y_e \\
\text{sub. to} \quad & \sum_{e:v \in e} y_e \leq w(v), \ \forall v \in V \\
& y_e \geq 0, \ \forall e \in E.
\end{aligned}
$$

### A.2 PSEUDOCODE OF MVC ALGORITHM

A $2/(1-\epsilon)$-approximation algorithm for the minimum vertex cover (MVC) problem was proposed by Khuller et al. (1994). It can be interpreted as an instantiation of the general primal-dual approximation algorithm without uniform increase (Algorithm 1). In the following, we give the original algorithm as illustrated in the original paper (Khuller et al., 1994).

Intuitively, the algorithm maintains a packing $p$ and partial cover $C_p = \{v \in V : p(E(v)) \geq (1-\epsilon)w(v)\}$, and gradually increases the edge weights $p(e)$ as much as possible. When the constraint on the residual vertex weight is met, a vertex $v$ is removed and added to the cover $C_p$. The process iterates until $p$ is $\epsilon$-maximal and $C_p$ is a cover. Let $E_p(v)$ be the set of remaining edges incident to vertex $v$, $d_p(v) = |E_p(v)|$ be the degree, and $w_p(v) = w(v) - p(E(v))$ be the residual weight. The following is a pseudocode of the algorithm as described in Khuller et al. (1994).

---

**Algorithm 2** COVER$(G = (V, E), w, \epsilon)$

---

1: **for** $v \in V$ **do**
2: $\quad w_p(v) \leftarrow w(v); E_p(v) \leftarrow E(v); d_p(v) \leftarrow |E(v)|$
3: **while** edges remain **do**
4: $\quad$ **for** each remaining edge $(u, v)$ **do**
5: $\quad\quad \delta((u, v)) \leftarrow \min(w_p(u)/d_p(u), w_p(v)/d_p(v))$
6: $\quad$ **for** each remaining vertex $v$ **do**
7: $\quad\quad w_p(v) \leftarrow w_p(v) - \sum_{e \in E_p(v)} \delta(e)$
8: $\quad\quad$ **if** $w_p(v) \leq \epsilon \, w(v)$ **then**
9: $\quad\quad\quad$ delete $v$ and its incident edges; update $E_p(\cdot)$ and $d_p(\cdot)$
$\quad$ **return** the set of deleted vertices

---

### A.3 PRIMAL-DUAL PAIR: SET COVER AND ELEMENT PACKING

The minimum set cover (MSC) problem is a generalization of MVC to hypergraphs. Similar to the edge packing, the element packing is the dual of the LP relaxation of the set cover problem.

**Definition 6** (Minimum Set Cover). *Given a universe $\mathcal{U}$ and a family of sets $\mathcal{C} \subseteq 2^{\mathcal{U}}$ with non-negative weights $w : \mathcal{C} \to \mathbb{R}^+$, a set cover is a subfamily $\mathcal{C}' \subseteq \mathcal{C}$ such that $\cup_{S \in \mathcal{C}'} S = \cup_{S \in \mathcal{C}} S$. The objective is to minimize the total weight $\sum_{S \in \mathcal{C}'} w(S)$.*

**Definition 7** (Maximum Element Packing). *An element-packing is an assignment $p : \mathcal{U} \to \mathbb{R}^+$ of non-negative weights to the elements $e \in \mathcal{U}$, such that for any set $S \in C$, the total weight $\sum_{e \in S} p(e)$ is at most $w(S)$. The objective is to maximize the total element weight $\sum_{e \in \mathcal{U}} p(e)$.*

Let $x_S \in \{0, 1\}$ indicate whether each set $S \in \mathcal{C}$ is in the cover $\mathcal{C}'$, and $y_e \in \mathbb{R}^+$ be the non-negative weight assigned to each element $e \in \mathcal{U}$. The two problems can be formulated as:

$$\text{Min} \quad \sum_{S \in \mathcal{C}} w(S) x_S \qquad\qquad \text{Max} \quad \sum_{e \in \mathcal{U}} y_e$$

$$\text{sub. to} \quad \sum_{S : e \in S} x_S \geq 1, \ \forall e \in \mathcal{U} \qquad\qquad \text{sub. to} \quad \sum_{e \in S} y_e \leq w(S), \ \forall S \in \mathcal{C}$$

$$x_S \in \{0, 1\}, \ \forall S \in \mathcal{C} \qquad\qquad y_e \geq 0, \ \forall e \in \mathcal{U}.$$

### A.4 PSEUDOCODE OF MSC ALGORITHM

The above algorithm can be extended for set cover as an $r/(1-\epsilon)$-approximation algorithm, where $r$ is the maximal cardinality of sets. The following pseudocode is the adapted approximation algorithm to solve vertex cover on a hypergraph.

---

**Algorithm 3** COVER($G = (V, E), w, \epsilon$)

---

1: **for** $v \in V$ **do**
2:      $w_p(v) \leftarrow w(v); E_p(v) \leftarrow E(v); d_p(v) \leftarrow |E(v)|$
3: **while** edges remain **do**
4:      **for** each remaining edge $e$ **do**
5:          $\delta(e) \leftarrow \min_{v \in e}(w_p(v)/d_p(v))$
6:      **for** each remaining vertex $v$ **do**
7:          $w_p(v) \leftarrow w_p(v) - \sum_{e \in E_p(v)} \delta(e)$
8:          **if** $w_p(v) \leq \epsilon \, w(v)$ **then**
9:              delete $v$ and its incident edges; update $E_p(\cdot)$ and $d_p(\cdot)$
     **return** the set of deleted vertices

---

## B PROOF OF THEOREM 1

**Theorem 1.** *Given a hitting set problem $(\mathcal{T}, E, w)$, let $\mathcal{A}(\mathcal{T}, E, w)$ be the solution produced by Algorithm 1, which terminates after $K$ timesteps. There exists a parameter configuration $\Theta$ for a PDGNN model $\mathcal{M}_\Theta$ such that, at timestep $K$, the model output satisfies $\mathcal{M}_\Theta^{(K)}(\mathcal{T}, E, w) = \mathcal{A}(\mathcal{T}, E, w)$. Furthermore, let $(\boldsymbol{x}^{(t)}, \boldsymbol{r}^{(t)}, \boldsymbol{\delta}^{(t)}, \Delta^{(t)})$ be the intermediate quantities computed by Algorithm 1 at each timestep $t$. Then, the PDGNN model satisfies $\mathcal{M}_\Theta^{(t)}(\mathcal{T}, E, w) = (\boldsymbol{x}^{(t)}, \boldsymbol{r}^{(t)}, \boldsymbol{\delta}^{(t)}, \Delta^{(t)})$.*

**Proof** Given a hitting set problem $(\mathcal{T}, E, w)$, we construct a bipartite graph $B$ as explained in Section 4.1. Let $B^{(t)}$ denote the bipartite graph at timestep $t$, then $B^{(0)} = B$. If we remove a node $e$ and its incident edges from the graph when an element $e$ is included in the hitting set at timestep $t$, an action denoted by $x_e^{(t)} = 1$, then the bipartite graph $B^{(t)}$ changes accordingly. Therefore, the degrees of primal nodes $\boldsymbol{d}^{(t)}$ can be computed via $\boldsymbol{d}^{(t)} = f(\mathcal{T}, E, \max_{t' \in [0,...,t]}(\boldsymbol{x}^{(t')}))$ for some function $f$, where the max function is applied element-wise. WLOG, let $n$ denote the hidden dimension. We can now prove the theorem using mathematical induction.

1. Base case ($t = 0$): This is true because the inputs $\mathcal{M}_\Theta^{(0)}(\mathcal{T}, E, w) = (\boldsymbol{0}, \{w_e : e \in E\}, \boldsymbol{0}, 0) = (\boldsymbol{x}^{(0)}, \boldsymbol{r}^{(0)}, \boldsymbol{\delta}^{(0)}, \Delta^{(0)})$.

2. Induction step ($t > 0$): To formulate the strong induction hypothesis, let $(\boldsymbol{x}^{(t')}, \boldsymbol{r}^{(t')}, \boldsymbol{\delta}^{(t')}, \Delta^{(t')})$ be the intermediate quantites computed by Algorithm 1 for each timestep $t' \in [0,...,t-1]$, assume $\mathcal{M}_\Theta^{(t')}(\mathcal{T}, E, w) = (\boldsymbol{x}^{(t')}, \boldsymbol{r}^{(t')}, \boldsymbol{\delta}^{(t')}, \Delta^{(t')})$. We now prove that $\mathcal{M}_\Theta^{(t)}(\mathcal{T}, E, w) = (\boldsymbol{x}^{(t)}, \boldsymbol{r}^{(t)}, \boldsymbol{\delta}^{(t)}, \Delta^{(t)})$.

The inputs for the $t^{\text{th}}$ step of our recurrent model are the outputs from the $(t-1)^{\text{th}}$ step. By the induction hypothesis, the model output $\hat{\boldsymbol{r}}^{(t-1)} = \boldsymbol{r}^{(t-1)}$. Furthermore, since $\boldsymbol{d}^{(t)} = f(\mathcal{T}, E, \max_{t' \in [0,...,t]}(\boldsymbol{x}^{(t')}))$ for some function $f$, and the model outputs satisfy $\hat{\boldsymbol{x}}^{(t')} = \boldsymbol{x}^{(t')}$ for all $t' \in [0,...,t-1]$ via the strong induction hypothesis, we have $\hat{\boldsymbol{d}}^{(t-1)} = \boldsymbol{d}^{(t-1)}$. We take the natural logarithmic form of $\boldsymbol{r}^{(t-1)}$ and $\boldsymbol{d}^{(t-1)}$ as inputs to the encoders.

Both encoders $f_r$ and $f_d$ are MLPs. We define the weight of $f_r$ as $W_{f_r} = [1, 0, ..., 0] \in \mathbb{R}^{n \times 1}$. We define the weight of $f_d$ as $W_{f_d} = [1, 0, ..., 0] \in \mathbb{R}^{n \times 1}$. Therefore,

$$\boldsymbol{h}_e^{(t)} = f_r(r_e^{(t-1)}) = W_{f_r} \ln r_e^{(t-1)} = [\ln r_e^{(t-1)}, 0, ..., 0] \in \mathbb{R}^{n \times 1}$$

$$\boldsymbol{h}_{d_e}^{(t)} = f_d(d_e^{(t-1)}) = W_{f_d} \ln d_e^{(t-1)} = [\ln d_e^{(t-1)}, 0, ..., 0] \in \mathbb{R}^{n \times 1}$$

The first step performs message-passing from the primal node representations $\boldsymbol{h}_e^{(t)}$ to the dual nodes $\boldsymbol{h}_T^{(t)}$ via $\boldsymbol{h}_T^{(t)} = \min_{e \in \mathcal{N}(T)} g_e(\boldsymbol{h}_e^{(t)}, \boldsymbol{h}_{d_e}^{(t)})$. Let $g_e$ be an MLP with ELU activation, then we set $W_{g_e} = [1, 0, ..., 0, -1, 0, ..., 0]^\top \in \mathbb{R}^{1 \times 2n}$, $b_{g_e} = [1] \in \mathbb{R}^1$, and $W'_{g_e} = [1, 0, ..., 0] \in \mathbb{R}^{n \times 1}$ thus:

$$g_e(\boldsymbol{h}_e^{\prime(t)}) = W'_{g_e} \left( \text{ELU} \left( W_{g_e} [\boldsymbol{h}_e^{(t)} \| \boldsymbol{h}_{d_e}^{(t)}] \right) + b_{g_e} \right)$$

$$= W'_{g_e} \left( \text{ELU} \left( [1, 0, ..., 0, -1, 0, ..., 0]^\top [\ln r_e^{(t-1)}, 0, ..., 0, \ln d_e^{(t-1)}, 0, ..., 0] \right) + 1 \right)$$

$$= W'_{g_e} \left( \text{ELU} \left( \ln r_e^{(t-1)} - \ln d_e^{(t-1)} \right) + 1 \right)$$

$$= W'_{g_e} \left( \text{ELU} \left( \ln \frac{r_e^{(t-1)}}{d_e^{(t-1)}} \right) + 1 \right)$$

Since $\text{ELU}(x) = e^x - 1$ if $x \leq 0$, and $\ln \frac{r_e^{(t-1)}}{d_e^{(t-1)}} \leq 0$,

$$g_e(\boldsymbol{h}_e^{\prime(t)}) = W'_{g_e} \left( \exp \left( \ln \frac{r_e^{(t-1)}}{d_e^{(t-1)}} \right) - 1 + 1 \right)$$

$$= [1, 0, ..., 0] \frac{r_e^{(t-1)}}{d_e^{(t-1)}}$$

$$= [\frac{r_e^{(t-1)}}{d_e^{(t-1)}}, 0, ..., 0] \in \mathbb{R}^{n \times 1}$$

Therefore, the hidden representation $\boldsymbol{h}_T^{(t)}$ for each set $T \in \mathcal{T}$, is:

$$\boldsymbol{h}_T^{(t)} = \min_{e \in \mathcal{N}(T)} g_e(\boldsymbol{h}_e^{\prime(t)})$$

$$= \min_{e \in \mathcal{N}(T)} [\frac{r_e^{(t-1)}}{d_e^{(t-1)}}, 0, ..., 0]$$

$$= [\min_{e \in \mathcal{N}(T)} \frac{r_e^{(t-1)}}{d_e^{(t-1)}}, 0, ..., 0]$$

$$= [\delta_T^{(t)}, 0, ..., 0] \in \mathbb{R}^{n \times 1} \quad \text{(From Line 5 of Algorithm 1)}$$

The next step performs message-passing from the dual node representations $\boldsymbol{h}_T^{(t)}$ and then updates the primal representations $\boldsymbol{h}_e^{(t)}$ via $\boldsymbol{h}_e^{(t)} = g_u(\boldsymbol{h}_e^{(t)}, \sum_{T \in \mathcal{N}(e)} \boldsymbol{h}_T^{(t)})$. We use ELU activation function on the previously computed $\boldsymbol{h}_e^{(t)}$ and a bias $b_r = [1, 0, ..., 0] \in \mathbb{R}^{n \times 1}$. Since $\ln r_e^{(t-1)} \leq 0$, we have:

$$\boldsymbol{h}_e^{(t)} = \text{ELU} \left( \boldsymbol{h}_e^{(t)} \right) + b_r$$

$$= \text{ELU} \left( [\ln r_e^{(t-1)}, 0, ..., 0] \right) + [1, 0, ..., 0]$$

$$= [\exp \left( \ln r_e^{(t-1)} \right) - 1 + 1, 0, ..., 0]$$

$$= [r_e^{(t-1)}, 0, ..., 0] \in \mathbb{R}^{n \times 1}$$

The update function $g_u$ is also an MLP. We define its weights to be $W_{g_u} = [1, 0, ..., 0, -1, 0, ..., 0]^\top \in \mathbb{R}^{1 \times 2n}$. Then, the hidden dimension $\boldsymbol{h}_e^{(t)}$ for each element $e \in E$ becomes:

$$
\begin{aligned}
\boldsymbol{h}_e^{(t)} &= g_u \left( \boldsymbol{h}_e^{(t)}, \sum_{T \in \mathcal{N}(e)} \boldsymbol{h}_T^{(t)} \right) \\
&= W_{g_u} \left[ \boldsymbol{h}_e^{(t)} \| \sum_{T \in \mathcal{N}(e)} \boldsymbol{h}_T^{(t)} \right] \\
&= [1, 0, ..., 0, -1, 0, ..., 0]^\top [r_e^{(t-1)}, 0, ..., 0, \sum_{T \in \mathcal{N}(e)} \delta_T^{(t)}, 0, ..., 0] \\
&= [r_e^{(t-1)} - \sum_{T \in \mathcal{N}(e)} \delta_T^{(t)}, 0, ..., 0] \\
&= [r_e^{(t)}, 0, ..., 0] \in \mathbb{R}^{n \times 1} \quad \text{(From Line 6 of Algorithm 1)}
\end{aligned}
$$

Alternatively, if the uniform increase rule is incorporated, we have an additional virtual node $z$ that connects to all dual variables. Its hidden representation $\boldsymbol{h}_z^{(t)}$ is computed as:

$$
\begin{aligned}
\boldsymbol{h}_z^{(t)} &= \min_{T \in \mathcal{T}} \boldsymbol{h}_T^{(t)} \\
&= \min_{T \in \mathcal{T}} [\delta_T^{(t)}, 0, ..., 0] \\
&= [\min_{T \in \mathcal{T}} \delta_T^{(t)}, 0, ..., 0] \\
&= [\Delta^{(t)}, 0, ..., 0] \in \mathbb{R}^{n \times 1} \quad \text{(From Line 6.1 of Algorithm 1)}
\end{aligned}
$$

Then let $\boldsymbol{h'}_T^{(t)} = \boldsymbol{h}_z^{(t)}$, the primal variable updates becomes:

$$
\begin{aligned}
\boldsymbol{h}_e^{(t)} &= g_u \left( \boldsymbol{h}_e^{(t)}, \sum_{T \in \mathcal{N}(e)} \boldsymbol{h'}_T^{(t)} \right) \\
&= W_{g_u} \left[ \boldsymbol{h}_e^{(t)} \| \sum_{T \in \mathcal{N}(e)} \boldsymbol{h'}_T^{(t)} \right] \\
&= [1, 0, ..., 0, -1, 0, ..., 0]^\top [r_e^{(t-1)}, 0, ..., 0, \sum_{T \in \mathcal{N}(e)} \Delta^{(t)}, 0, ..., 0] \\
&= [r_e^{(t-1)} - d_e^{(t-1)} \Delta^{(t)}, 0, ..., 0] \\
&= [r_e^{(t)}, 0, ..., 0] \in \mathbb{R}^{n \times 1} \quad \text{(From Line 6.2 of Algorithm 1)}
\end{aligned}
$$

For the decoders $q_x$, $q_r$, $q_\delta$ (and $q_\Delta$ if uniform increase is used), they map hidden representations $\boldsymbol{h}_e^{(t)}$, $\boldsymbol{h}_T^{(t)}$ (and $\boldsymbol{h}_z^{(t)}$ if uniform increase is used) to predictions for the intermediate quantities computed by the algorithm. We define $W_{q_x} = W_{q_r} = W_{q_\delta} = W_{q_\Delta} = [1, 0, ..., 0]^\top \in \mathbb{R}^{1 \times n}$. For $x_e^{(t)}$, it is a binary classification task, where $x_e^{(t)} = 1$ if $r_e^{(t)} = 0$ to add the element $e$ to the hitting set. Define $o : \mathbb{R} \to \{0, 1\}$, where

$$
o(x) = \begin{cases} 1, & \text{if } x \leq 0 \\ 0, & \text{else} \end{cases}
$$

We note that although we use a sigmoid function in our architecture, the sigmoid function can approximate $o(x)$ to arbitrary precision by adjusting its temperature. Therefore, we have

$$
\begin{aligned}
\hat{x}_e^{(t)} &= o(q_x(\boldsymbol{h}_e^{(t)})) \\
&= o \left( W_{q_x}(\boldsymbol{h}_e^{(t)}) \right)
\end{aligned}
$$

$$= o\left([1,0,...,0]^\top [r_e^{(t)}, 0, ..., 0]\right)$$

$$= o\left(r_e^{(t)}\right)$$

$$= x_e^{(t)} \quad \text{(Definition of } x_e^{(t)})$$

For the other three intermediate quantities:

$$\hat{r}_e^{(t)} = q_r(\boldsymbol{h}_e^{(t)})$$

$$= W_{q_r}(\boldsymbol{h}_e^{(t)})$$

$$= [1,0,...,0]^\top [r_e^{(t)}, 0, ..., 0]$$

$$= r_e^{(t)}$$

$$\hat{\delta}_T^{(t)} = q_\delta(\boldsymbol{h}_T^{(t)})$$

$$= W_{q_\delta}(\boldsymbol{h}_T^{(t)})$$

$$= [1,0,...,0]^\top [\delta_T^{(t)}, 0, ..., 0]$$

$$= \delta_T^{(t)}$$

$$\hat{\Delta}^{(t)} = q_\delta(\boldsymbol{h}_z^{(t)})$$

$$= W_{q_\Delta}(\boldsymbol{h}_z^{(t)})$$

$$= [1,0,...,0]^\top [\Delta^{(t)}, 0, ..., 0]$$

$$= \Delta^{(t)}$$

Therefore, $\Theta_{\text{dual}}^{(t)}(\mathcal{T}, E, w) = (\boldsymbol{x}^{(t)}, \boldsymbol{r}^{(t)}, \boldsymbol{\delta}^{(t)}, \Delta^{(t)})$ and the induction step is completed.

## C  DATASETS

### C.1  SYNTHETIC DATASETS

We provide the details of the graph distributions used to generate random graphs for both training and testing.

**Bipartite graph**  To generate random bipartite graphs, given the number of nodes in the bipartite graph, the number of nodes on the two sides is randomly chosen, with at least two nodes on each side. Two nodes from each side are connected with a probability of 0.2. Node weights for primal variables are uniformly sampled from $[0, 1]$. We ensure that each bipartite graph is connected.

**Barabási-Albert (B-A) graph**  Barabási-Albert graphs for both training and testing are randomly generated using `networkx.barabasi_albert_graph`. The number of edges to attach from a new node to existing nodes is randomly chosen from $[1, 10]$. Node weights for primal variables are uniformly sampled from $[0, 1]$.

**Erdős–Rényi (E-R) graph**  We use `networkx.erdos_renyi_graph` to generate random Erdős–Rényi graphs with edge probability uniformly sampled from $[0.2, 0.8]$.

**Star graph**  Star graphs are generated by randomly partitioning the nodes into 1 to 5 sets. Within each node set, a star graph is generated with a center node connected to all other nodes. Random edges between the star graphs are then added to ensure the graph is connected.

**Lobster graph**  To generate a lobster graph, the number of nodes on the "backbone" $m$ is randomly sampled from $[1, n-1]$, where $n$ is the total number of nodes in the graph. Then, another $k$ nodes are added to the backbone nodes to start "branches", where $1 \le k \le (n-m)$. Finally, the remaining nodes (if any left from $n - m - k$) are randomly attached to the branches.

**3-Connected (3-Con) Planar graph** A 3-regular graph is randomly generated with `networkx.random_regular_graph`, and then checked to see if it is 3-connected and planar. The process is repeated until a valid 3-connected planar graph is found, or if it reaches the limit of 100 attempts.

Table 5: Percentage of nodes being in the optimal vertex cover for different graph families.

|          | B-A | E-R | Star | Lobster | 3-Con |
|----------|-----|-----|------|---------|-------|
| 16 nodes | 43% | 71% | 23%  | 40%     | 61%   |
| 32 nodes | 49% | 80% | 15%  | 41%     | 60%   |
| 64 nodes | 45% | 88% | 8%   | 41%     | 59%   |

For vertex cover and set cover, we use Algorithm 2 and Algorithm 3 (Khuller et al., 1994) to generate intermediate supervisions, which are instantiations of Algorithm 1 with improved efficiency, as explained in Section 3.2. For hitting set, we use Algorithm 1 with the uniform increase rule. Furthermore, the optimal solutions are generated with the default IP solver in `scipy`, which is based on HiGHS (Schwendinger & Schumacher, 2023; Huangfu & Hall, 2018).

## C.2 GUROBI DATASETS

Random B-A graphs are generated following the same distribution described above. For testing, we generate B-A graphs with 500, 600, and 750 nodes. We use the trained model to perform inference on the testing set and retrieve vertex cover solutions. We also use Algorithm 2 (Khuller et al., 1994) to compute solutions with $\epsilon = 0.1$. The comparison of the solutions from the model and the algorithm is shown in Table 6. The two sets of solutions are then used to initialize variables for warm starting the Gurobi solver. We also compare them with the default initialization (i.e. no warm start) of Gurobi. We use the default parameter settings of Gurobi, setting thread count to 1 (`model.setParam('Threads', 1)`), time limit to 3600s (`model.setParam('TimeLimit', 3600)`), and random seed (`model.setParam('Seed', seed)`). Each experiment is repeated with 5 seeds.

Table 6: The ratio of total weights of the solutions generated by the model compared with those generated by the algorithm ($w_{\text{pred}}/w_{\text{algo}}$). Comparison with the optimal solutions is not included due to the large size of graphs. We also report the percentage of uncovered edges from the solutions generated by the model (i.e. how often the cleanup stage is required).

|                                  | 500 nodes | 600 nodes | 750 nodes |
|----------------------------------|-----------|-----------|-----------|
| $w_{\text{pred}}/w_{\text{algo}}$ | 0.972     | 0.970     | 0.970     |
| Uncovered edges                  | 0%        | 0%        | 0%        |

## C.3 REAL-WORLD DATASETS

**Airports** The Airports datasets (Ribeiro et al., 2017) consist of three airport networks from Brazil (131 nodes, 1038 edges), Europe (399 nodes, 5995 edges), and the USA (1190 nodes, 13599 edges). The nodes represent airports, and the edges represent commercial flight routes. The node features are one-hot encoded node identifiers, as described in Jin et al. (2019). The task is to predict the activity level of each airport, measured by the total number of landings plus takeoffs, or the total number of people arriving plus departing. It is a classification task with 4 labels, with label 1 assigned to the 25% least active airports, and so on, according to the quartiles of the activity distribution. We create 10 random train/val/test splits for the transductive task with a ratio of $60\%/20\%/20\%$.

**WikipediaNetwork** We use the preprocessed WikipediaNetwork datasets (Pei et al., 2020) from the original datasets (Rozemberczki et al., 2019). It consists of two datasets, Chameleon (2277 nodes, 36101 edges, 2325 features) and Squirrel (5201 nodes, 217073 edges, 2089 features). The nodes are web pages and edges represent hyperlinks between them. The task is to predict the average monthly traffic of the web page, which is categorized into five classes for prediction. We use the 10 provided train/val/test splits from Pei et al. (2020) with ratio $60\%/20\%/20\%$ for this transductive task.

**PPI** The PPI dataset (Zitnik & Leskovec, 2017) consists of 20 protein-protein interaction networks. On average, each graph has 2245.3 nodes and 61318.4 edges. Each node has 50 features (such as positional gene sets, motif gene sets, immunological signatures) with 121 labels taken from the gene ontology sets. This inductive dataset provides splits for train/val/test sets. Each experiment is repeated with 10 seeds.

# D  HYPERPARAMETERS

All GPU experiments were performed on Nvidia Quadro RTX 8000 with 48GB memory. The Gurobi experiments were conducted on Intel Xeon E7-8890x with 144 cores and 12TB memory.

**Synthetic and Gurobi experiments** For training, we use the Adam optimizer with an initial learning rate of 1e-3 and weight decay of 1e-4, coupled with the ReduceLROnPlateau scheduler with default settings. Additionally, we use a batch size of 32, a hidden dimension of 32, and a maximum of 100 epochs. For testing, we use the trained model with the lowest validation loss.

**Real-world dataset experiments** We use the same optimizer and scheduler settings as in the synthetic experiments. Additionally, we also apply early stopping with a patience of 10 epochs based on validation loss and set the scheduler with a patience of 20 epochs. All embeddings have a fixed dimension of 32. For testing, we use the model with the lowest validation loss. The base models are used with max jumping knowledge (Xu et al., 2018) and L2 normalization after each layer (Rossi et al., 2024). We conduct hyperparameter search for each base model on Airports datasets (Brazil, Europe, USA), then use the setting for all embedding methods. Due to the high computational costs of WikipediaNetwork (Chameleon, Squirrel) and PPI datasets, hyperparameter search is only done with GCN. We use the default TPE hyperparameter search algorithm from `optuna` (Akiba et al., 2019) with a median pruner. The searchable parameters are lr=[0.01, 0.001, 0.0005], hid_dim=[32, 64, 128], dropout=[0.1, 0.3, 0.5], and num_layer=[1, 3, 5]. For training Node2Vec (Grover & Leskovec, 2016), we use walk_length=20, context_size=10, walks_per_node=10, with 100 epochs.

Table 7: Additional hyperparameters for real-world dataset experiments.

|  | Brazil | Europe | USA | Chameleon | Squirrel | PPI |
|---|---|---|---|---|---|---|
| GCN | lr=0.0005
hid_dim=32
dropout=0.5
num_layer=3 | lr=0.001
hid_dim=32
dropout=0.1
num_layer=3 | lr=0.001
hid_dim=64
dropout=0.3
num_layer=3 | lr=0.001
hid_dim=128
dropout=0.3
num_layer=3 | lr=0.001
hid_dim=128
dropout=0.3
num_layer=3 | lr=0.01
hid_dim=32
dropout=0.1
num_layer=3 |
| GAT | lr=0.001
hid_dim=32
dropout=0.3
num_layer=3 | lr=0.001
hid_dim=32
dropout=0.1
num_layer=3 | lr=0.001
hid_dim=32
dropout=0.1
num_layer=3 | as above | as above | as above |
| SAGE | lr=0.0005
hid_dim=32
dropout=0.5
num_layer=1 | lr=0.0005
hid_dim=32
dropout=0.3
num_layer=1 | lr=0.001
hid_dim=64
dropout=0.1
num_layer=3 | as above | as above | as above |

# E  ADDITIONAL ARCHITECTURAL DETAILS

In practice, due to the potentially high degree variance, we apply log transformation on the node degree $d_e^{(t)}$ before encoding it with $f_d$ for better generalization, i.e. $\boldsymbol{h}_{d_e}^{(t)} = f_d(\ln(d_e^{(t-1)} + 1))$. Then, for decoding $\hat{x}_e^{(t)} = q_x(\boldsymbol{h}_e^{(t)})$, which represents whether to include element $e$ to the solution, we apply a sigmoid activation function to convert logits to probabilities. For minimum vertex cover and minimum set cover, since multiple elements can be included into the solution at each timestep, we set a threshold of 0.5 to decide whether to include the element. For minimum hitting set, since the uniform increase rule is used, only one element is included into the solution at each timestep, we choose the element with the highest probability to include in the hitting set. Lastly, we add dropouts

in between processor layers with probability of 0.2 for Gurobi and real-world experiments. This helps the model to generalize to much larger graphs at a slight cost of approximation ratios.

## F    LIMITATION AND FUTURE WORK

Our framework can be adapted to solve a wide range of combinatorial optimization problems that can be represented using the hitting set formulation. The hitting set provides a flexible structure for modeling various combinatorial problems by selecting a subset of elements that "hits" or "covers" all required constraints, represented as sets. This formulation is versatile because it can capture diverse constraints (e.g., nodes, edges, paths, cycles), making it applicable to numerous optimization problems. As discussed in Section 3.2 , Algorithm 1 can be reformulated to recover many classical (exact or approximation) algorithms for problems that are special cases of the hitting set, covering both polynomial-time solvable and NP-hard problems. Some of these special cases are illustrated in Goemans & Williamson (1996) and Williamson & Shmoys (2011), including shortest s-t path, minimum spanning tree, vertex cover, set cover, minimum-cost arborescence, feedback vertex set, generalized Steiner tree, minimum knapsack and facility location problems.

We note that not all problems can be directly represented by the hitting set formulation. As a minimization problem, the hitting set does not naturally align with maximization objectives, making the primal-dual approximation algorithm less straightforward to apply. However, the primal-dual framework can still be extended to maximization problems by carefully reformulating the primal-dual pair, where the dual is a minimization problem, and adapting Algorithm 1. Then, similarly, the algorithm starts with a feasible dual solution and iteratively updates both primal and dual variables to reduce the gap between them. Therefore, PDGNN, with its bipartite graph structure, can still be extended to handle maximization problems with appropriate adjustments to Algorithm 1.

Furthermore, while Algorithm 1 provides a general framework for designing primal-dual approximation algorithms, it can be further strengthened by incorporating techniques to enhance the algorithmic performance. One such technique is the uniform increase rule, which we have shown how it can be integrated into our framework. Future work can incorporate other advanced techniques, such as those outlined in Williamson & Shmoys (2011), to further extend our framework's ability to accommodate a broader range of primal-dual approximation algorithms with improved worst-case guarantees.

Lastly, our method may not explicitly preserve the worst-case approximation guarantees of the primal-dual algorithm in practice. Instead, our model focuses on learning solutions that perform better for the training distribution and generalize well to new instances. While this does not mean that the worst-case guarantees are secured, it is important to note that such cases are often less common in real-world scenarios. One of the core strengths of NAR lies in leveraging a pretrained GNN with embedded algorithmic knowledge to tackle real-world datasets. Thus, we believe it is valuable to train models to produce high-quality solutions for common cases, even if they do not preserve worst-case guarantees. This aligns with the overarching goals of NAR.

## G    RELATED WORKS ON NEURAL ALGORITHMIC REASONING

Lastly, we provide a more comprehensive review of existing works on Neural Algorithmic Reasoning (NAR) and highlight our contributions in this context.

**Neural algorithmic reasoning**    The algorithmic alignment framework proposed by Xu et al. (2020) suggests that GNNs are particularly well-suited for learning dynamic programming algorithms due to their shared aggregate-update mechanism. Additionally, Veličković et al. (2020) demonstrates the effectiveness of GNNs in learning graph algorithms such as BFS and Bellman-Ford. These foundational works have contributed to the development of neural algorithmic reasoning (Veličković & Blundell, 2021), which investigates the potential of neural networks, particularly GNNs, to simulate traditional algorithmic processes. This research direction has since inspired several follow-up studies, including efforts to instantiate the framework for specific algorithms (Deac et al., 2020; Georgiev & Liò, 2020; Zhu et al., 2021), applications to real-world use cases (Deac et al., 2021; He et al., 2022; Beurer-Kellner et al., 2022; Georgiev et al., 2023a; Numeroso et al., 2023; Estermann et al., 2024), architectural improvements (Georgiev et al., 2022; Bevilacqua et al., 2023; Rodionov & Prokhorenkova, 2023; Jain et al., 2023; Engelmayer et al., 2023; Mirjanic et al., 2023; Georgiev et al.,

2023b; Dudzik et al., 2024; Jürß et al., 2024; Xhonneux et al., 2024; Georgiev et al., 2024; Rodionov & Prokhorenkova, 2024; Xu & Veličković, 2024; Kujawa et al., 2024), and integration with large language models (Bounsi et al., 2024). Our work advances NAR by introducing a general framework designed to tackle combinatorial optimization problems, particularly NP-hard ones, with the objective of simulating and outperforming primal-dual approximation algorithms.

**Combinatorial optimization with GNNs**   The CLRS benchmark (Veličković et al., 2022) and its extensions (Minder et al., 2023; Markeeva et al., 2024) are widely recognized for evaluating GNNs on 30 algorithms from the CLRS textbook (Cormen et al., 2001) and more. However, these algorithms are limited to polynomial-time problems, leaving the more challenging NP-hard problems largely unexplored in neural algorithmic reasoning. A comprehensive review by Cappart et al. (2022) summarizes the current progress of using GNNs for combinatorial optimization. The most relevant work (Georgiev et al., 2023c) trains GNNs on algorithms for polynomial-time-solvable problems and test them on NP-hard problems, demonstrating the value of algorithmic knowledge over non-algorithmically informed models. In contrast, our approach bridges this gap by extending GNNs to tackle NP-hard problems through the use of primal-dual approximation algorithms. Furthermore, we integrate optimal solutions from integer programming, which guides the model toward better outcomes during training. To the best of our knowledge, our method is the first of its kind to surpass the performance of the algorithms it was originally trained on.

**Multi-task learning for NAR**   Early work by Veličković et al. (2020) demonstrated that BFS and Bellman-Ford are best learned jointly, and subsequent studies have highlighted broader benefits of multi-task learning when GNNs are trained on multiple algorithms simultaneously (Xhonneux et al., 2021; Ibarz et al., 2022). Building on this, Numeroso et al. (2023) leveraged the primal-dual principle from linear programming to successfully learn the Ford-Fulkerson algorithm using the max-flow min-cut theorem. However, their approach was tailored specifically for Ford-Fulkerson and did not generalize to other primal-dual scenarios or address NP-hard problems. To overcome these limitations, our work introduces a general framework that employs the primal-dual principle to enable GNNs to benefit from multi-task learning across a broad range of optimization problems, particularly those expressible as instances of the general hitting set problem.

# H    CONNECTION WITH NEURAL COMBINATORIAL OPTIMIZATION (NCO)

While NCO is not the primary focus of our work, our method also contributes to an underexplord area of NCO. Specifically, we propose an algorithmically informed GNN that effectively addresses critical challenges of data efficiency and generalization.

Most GNN-based supervised learning methods for NCO learn task-specific heuristics or optimize solutions in an end-to-end manner (Joshi et al., 2019; Li et al., 2018; Gasse et al., 2019b; Fu et al., 2021). These end-to-end approaches rely exclusively on supervision signals derived from optimal solutions, which are computationally expensive to obtain for hard instances. Furthermore, dependence on such labels can limit generalization (Joshi et al., 2022). In contrast, our method trains on synthetic data obtained efficiently from a polynomial-time approximation algorithm. The embedding of algorithmic knowledge also demonstrates strong generalization. The addition of optimal solutions are derived from small problem instances, enabling our model to outperform the approximation algorithm and generalize effectively to larger problem sizes.

Our approach represents a previously underexplored area of NCO research with GNNs, offering a general and effective framework. Unlike end-to-end methods, we leverage intermediate supervision signals from polynomial-time approximation algorithms, which can be generated efficiently, to address key bottlenecks in data efficiency and generalization. Additionally, we fill the gap in autoregressive methods for NCO using GNNs by aligning our architecture with the primal-dual method, enabling the GNN to simulate a single algorithmic step in an efficient and structured manner.

## I    DUALITY IN LINEAR PROGRAMMING

Duality in linear programming has been utilized in training neural networks to tackle combinatorial optimization problems. Also for NAR, Numeroso et al. (2023) studied the polynomial-time-solvable max-flow problem. They trained a GNN to imitate Ford-Fulkerson, an algorithm built on the connections between the max-flow problem and its linear programming dual, the min-cut problem. Thus, both our paper and the paper by Numeroso et al. (2023) leverage the notion of *duality*, which is a essential tool in algorithm design. Numeroso et al.'s architecture was specialized for the Ford-Fulkerson algorithm and is not applicable to other problems (such as the NP-hard problems we focus on). We require a completely different approach because although Ford-Fulkerson and the primal-dual method for NP-hard problems are based on duality, one is not a special case of the other; they are fundamentally different algorithms. We present a general framework applicable to several different NP-hard problems.

Furthermore, Li et al. (2024) introduces a Learning-to-Optimize method to mimic Primal-Dual Hybrid Gradient method for solving large-scale LPs. While they focus on developing efficient solvers for LPs, we aim to simulate the primal-dual approximation algorithm for NP-hard problems using GNNs. Although both approaches reference the primal-dual framework, this similarity is superficial. The primal-dual terminology is widely used in optimization, but our work applies it to study algorithmic reasoning. For example, the primal-dual approximation algorithm can be instantiated to many traditional algorithms, such as Kruskal's algorithm for MST. Furthermore, unlike Li et al. (2024), our method relies on intermediate supervision from the primal-dual algorithm to guide reasoning, ensuring that the model learns to mimic algorithmic steps. Additionally, we incorporate optimal solutions into the training process to improve solution quality, allowing our model to outperform the primal-dual algorithm it is trained on. Moreover, the architectures differ significantly: our method employs a recurrent application of a GNN to iteratively solve problems, while Li et al. (2024) does not use GNNs or recurrent modeling. These distinctions highlight that our focus is not on solving LPs but on leveraging NAR to generalize algorithmic reasoning for NP-hard problems.

