# OpenReview forum: "Primal-Dual Graph Neural Networks for General NP-Hard Combinatorial Optimization"
_ICLR.cc/2025/Conference — Submitted to ICLR 2025_

### Official Review · Reviewer_EkCx · 2024-10-17

**Soundness:** 3
**Presentation:** 3
**Contribution:** 2
**Rating:** 5
**Confidence:** 4

**Summary:**

This paper propose a new GNN architecture, termed PDGNN, for solving the integer programming $\min \vec{w}^T\cdot\vec{x}$ s.t. $A\vec{x}\geq \vec{1}$ and $x_i\in \{0,1\}$, which can model many combinatorial optimization problems, such as set cover and hitting set. This GNN is designed by unrolling a primal-dual approximation algorithm. Numerical experiments are conducted to evaluate the performance of PDGNN.

**Strengths:**

The observation that the primal-dual algorithm naturally align with message-passing GNN is interesting. The integer programming studied in this paper is a quite general problem. The writing is clear and easy to follow.

**Weaknesses:**

1. About Section 1.1: The phenomenon that GNNs designed by unrolling an algorithm surpass the performance of the algorithm itself is not surprising (since the algorithm is an instantiation of the GNN, so the GNN has enough expressive power to surpass). In fact, the phenomenon is general: see. e.g.  "PDHG-unrolled learning-to-optimize method for large-scale linear programming" in ICML 2024. As in
 real-world scenarios, PDGNN will commonly be used as a warm start, so the paper could highlight how much computational time can be saved by employing PDGNN  as a warm-start.

2. As shown in Table 3, by employing PDGNN  as a warm-start, the commercial solver achieve a $\leq 1.1\times$ speedup compared to the vanilla version. The improvement did not impress me, as I had been expecting a $\geq 2\times$ speedup.

3. The paper proves that PDGNN can exactly replicates the primal-dual algorithm, but lacks explicit theoretical guarantees on the efficiency and effectiveness of PDGNN. For example, I would like to see theorems like "PDGNN  of xx layers has enough expressive power to output an xx-approximate solution for xxx problems"，or discussions on the challenges in deriving such theoretical guarantees or intuitions about what kind of guarantees might be possible given the current framework.

**Questions:**

1. Refer to "Weaknesses".

2. In the experiments, how many layers does PDGNN has?

---

> ### Author Response · Authors · 2024-11-21
> **Rebuttal (Part 1)**
>
> We are sincerely grateful to Reviewer EkCx for their kind feedback, noting that our method is interesting, our framework is general, and our writing is clear and easy to follow. We also greatly appreciate their thoughtful concerns and questions, which we have put into thorough consideration. The manuscript has been revised accordingly, with changes highlighted in blue. Below, we provide detailed responses to the concerns raised, offering additional context and a comprehensive discussion of our method's contributions and evaluation.
>
> **1. Unrolling the algorithm:**
>
> We agree with Reviewer EkCx that once it is established that a GNN can represent the unrolled steps of an algorithm, it becomes natural to hope that the GNN's expressive power might enable it to surpass the algorithm's performance. However, demonstrating that a GNN can represent the primal-dual algorithm for NP-hard problems is far from straightforward. GNNs are inherently designed for local message passing, and it is not immediately evident that this mechanism alone is sufficient to produce high-quality solutions to complex, combinatorial problems. A key contribution of this work is designing an architecture that enables the transmission of information across distant nodes (e.g. virtual node that connects all dual variables), ensuring alignment with the primal-dual algorithm, while avoiding the pitfall of over-squashing—a common challenge in GNNs. Furthermore, we demonstrate the advantage of incorporating optimal solutions alongside unrolling, enabling the GNN to surpass the performance of the algorithm it emulates, even when generalizing to larger problem instances. This highlights a key strength of our architecture.
>
> We also want to clarify the differences with [1]. We direct Reviewer EkCx to **General Responses** where we provide more context on Neural Algorithmic Reasoning (NAR), which is our main focus. Specifically, [1] introduces a Learning-to-Optimize method to mimic first-order methods for solving large-scale LPs. While [1] focuses on developing efficient solvers for LPs, we aim to simulate the primal-dual approximation algorithm for NP-hard problems using GNNs. Although both approaches reference the primal-dual terminology, this similarity is not straightforward. Duality is widely used in optimization, but our work applies it to study algorithmic reasoning. For example, the primal-dual approximation algorithm can be instantiated to many traditional algorithms, such as Kruskal’s algorithm for MST. Furthermore, unlike [1], our method relies on intermediate supervision from the primal-dual algorithm to guide reasoning, ensuring that the model learns to mimic algorithmic steps. Additionally, we incorporate optimal solutions into the training process to improve solution quality. Moreover, the architectures differ significantly: our method employs a recurrent application of a GNN to iteratively solve problems, while [1] does not use GNNs or recurrent modeling. These distinctions highlight that our focus is not on solving LPs but on leveraging NAR to generalize algorithmic reasoning for NP-hard problems. We have added a discussion on duality in LP in Appendix I of our revised manuscript. We sincerely thank Reviewer EkCx for raising this question, which provided us with an opportunity to better clarify our goals.

---

> > ### Author Response · Authors · 2024-11-21
> > **Rebuttal (Part 2)**
> >
> > **2. Warm-start speedup:**
> >
> > We value the concern pointed out by Reviewer EkCx on the speedup brought by PDGNN’s warm-starts. We politely point out that achieving performance improvements over state-of-the-art solvers like CPLEX and Gurobi is widely recognized as an extremely challenging task [2]. Thus, even a 10% improvement is noteworthy. As demonstrated in Table 6, our model exhibits strong generalization, consistently outperforming the approximation algorithm on graphs of size 750, despite being trained only on graphs of size 16. Additionally, the reduction in search time increases as graph sizes grow, while inference time remains constant. This suggests significant potential for our model to deliver even greater speedups as problem instances scale or when trained on larger graphs.
> >
> > Beyond warm-starting solvers, we also highlight another important application of our model, as detailed in Section 5.4. A key strength of NAR is its ability to embed algorithmic knowledge into neural architectures, making it well-suited for tasks requiring structured reasoning. For example, a model pre-trained with Bellman-Ford algorithm can apply to real-world routing and transportation challenges. In our case, since NAR employs an encoder-processor-decoder framework, we show that a GNN processor pretrained with primal-dual algorithm, combined with a new encoder, can directly process real-world datasets without requiring feature engineering to conform to rigid input formats of traditional algorithms. This produces embeddings that effectively capture algorithmic knowledge, improving performance on node-classification tasks in real-world applications (Table 4). These real-world applications, also demonstrated by [3][4], highlight the importance of extending NAR to the NP-hard domain, overcoming the limitations of polynomial-time-solvable problems, and enabling the resolution of more complex and impactful challenges.
> >
> > **3. Theoretical guarantee on efficiency and effectiveness:**
> >
> > We thank Reviewer EkCx for raising the question regarding our theoretical guarantees on the efficiency and effectiveness of PDGNN. In Theorem 1, we proved that there exists an instantiation of PDGNN capable of exactly replicating the primal-dual algorithm. Specifically, in this setup, the two encoders and three decoders are all 1-layer MLPs, the first component of the processor has two layers, and the second component has one layer, resulting in a total of 8 layers. This result demonstrates that an 8-layer PDGNN can replicate a single step of the primal-dual algorithm exactly. Consequently, PDGNN inherits the approximation ratio and convergence guarantees of the original algorithm it learns. For instance, in the case of MVC, we train PDGNN with the primal-dual approximation algorithm proposed by [3], which achieves a $2/(1-\epsilon)$ approximation ratio and converges in $K = O(\log m \log \frac{1}{\epsilon})$ iterations. Thus, we can extend Theorem 1 to state “For any $\epsilon$ between 0 and 1, there exists a parameterization $M_{\Theta}$ of PDGNN with 8 layers such that $M_{\Theta}^{(K)}$ yields a $2/(1-\epsilon)$-approximation to the optimal solution, where $K = O(\log m \log \frac{1}{\epsilon})$”. We will incorporate the extended theorem into our paper and provide additional discussion in Appendix B to elaborate on its approximation ratio and convergence guarantees. We sincerely thank Reviewer EkCx for raising this important question, which has allowed us to significantly enhance our theoretical analysis and better emphasize the efficiency and effectiveness guarantees of our approach.
> >
> > **Answers to questions:**
> > - Q: How many layers does PDGNN have?
> > - A: In practice, we use a 9-layer PDGNN, with both components of the processor implemented as two-layer MLPs. Additionally, PDGNN is applied in a recurrent manner, terminating either when a valid solution is reached or after a maximal number of iterations. The maximum number of iterations is set equal to the number of primal variables.

---

> > > ### Author Response · Authors · 2024-11-21
> > > **Rebuttal (Part 3)**
> > >
> > > We deeply appreciate Reviewer EkCx for recognizing our method as interesting, our framework as general, and our writing as clear and easy to follow. We are also grateful for the constructive suggestions and insightful clarification questions, which have been instrumental in enhancing both our theoretical proofs and the presentation of our contributions. We hope that the revised proofs and explanations address your concerns and provide greater clarity regarding the strengths of our work. We look forward to hearing your feedback and are happy to address any additional questions to ensure our contributions are fully understood and effectively conveyed!
> > >
> > > [1] Li & Yang et al., PDHG-Unrolled Learning-to-Optimize Method for Large-Scale Linear Programming, ICML 2024.
> > >
> > > [2] Nair et al., Solving Mixed Integer Programs Using Neural Networks, 2021.
> > >
> > > [3] Veličković and Blundell, Neural Algorithmic Reasoning, Patterns, 2021.
> > >
> > > [4] Numeroso et al., Dual Algorithmic Reasoning, ICLR 2023.

---

> > > > ### Comment · Reviewer_EkCx · 2024-11-22
> > > >
> > > > Thank you for the detailed responses and clarification of the contributions.
> > > >
> > > > I've an additional question: What advantages does your NAR offer compared to directly hardcoding the primal-dual algorithm into the network?

---

> ### Author Response · Authors · 2024-11-23
> **Reply to additional question**
>
> Thank you for the follow-up question! The primary advantage of NAR over hardcoding the primal-dual algorithm lies in the ability of neural networks to operate in a high-dimensional latent space, equipping the model with more expressive power than merely replicating the algorithm.
>
> - Firstly, our approach leverages optimal solutions as additional training signals. This means that our goal is not to exactly replicate the primal-dual approximation algorithm but to surpass its performance through additional supervision from optimal solutions of small instances, as demonstrated in Tables 1 and 2. This level of performance enhancement is not achievable by hardcoding the algorithm. Instead, our method simulates the algorithm using learnable functions in a high-dimensional space, which also improves generalization, maintaining the superior performance against the algorithm, to larger problem instances.
>
> - Secondly, as outlined in Section 5.4, one of NAR’s key motivations is to equip models with algorithmic reasoning capabilities, enabling them to tackle real-world tasks requiring structured reasoning. The NAR framework comprises an encoder, a GNN-based processor, and a decoder. During the pretraining stage, the encoder maps scalar inputs (e.g., weights) into a high-dimensional latent space, where the GNN simulates the algorithm (in our case, the primal-dual algorithm). The decoder then maps this output back to produce solutions. This pretrained GNN, imbued with algorithmic knowledge, can subsequently be applied to real-world datasets by replacing the encoder and decoder with components that map node features to and from the shared high-dimensional latent space. In contrast, hardcoding the algorithm would require reducing node features to scalar values, severely limiting the expressive power of the model. Furthermore, NAR learns to approximate the algorithm and can handle noise or incomplete data, accounting for uncertainty or patterns in real-world data. Therefore, NAR has the advantage to combine the benefits from flexibility of deep learning and structured reasoning of traditional algorithms. This use case is shown in Table 4 and prior NAR works (e.g. [3][4]). An illustrative example is Figure 1 of [3]. It also further motivates our extension of NAR to the NP-hard domain, given the prevalence of NP-hard problems in real-world applications.
>
> We hope this addresses your question and highlights the advantages of NAR. Please let us know if you have any additional questions so we can ensure all your concerns are properly addressed.
>
> [3] Veličković and Blundell, Neural Algorithmic Reasoning, Patterns, 2021.
>
> [4] Numeroso et al., Dual Algorithmic Reasoning, ICLR 2023.

---

> > ### Author Response · Authors · 2024-11-26
> >
> > Dear Reviewer EkCx,
> >
> > Hope this note finds you well! We're just checking in since it is almost the end of the rebuttal period and we were wondering if we have addressed your main concerns.
> >
> > To recap, we believe the main concerns are:
> >
> > - **Unrolling the algorithm**: We explained that representing the primal-dual algorithm for NP-hard problems with a GNN is far from straightforward, and highlighted the strength of our model design. We also clarified our differences with [1].
> > - **Warm-start speedup**: We pointed out that achieving performance improvements over Gurobi is widely recognized as challenging, making our improvement noteworthy. We also highlighted the other important practical usage of our NAR model - pretraining a GNN with algorithmic knowledge and applying it to real-world problems.
> > - **Theoretical guarantees**: Following Reviewer EkCs’s suggestion, we improved our theoretical analysis to incorporate the efficiency and approximation guarantees and number of layers involved.
> > - **Advantages of NAR**: We further elaborated on the two main advantages of NAR over hardcoding the algorithm within the model. Instead of exactly replicating, NAR leverages the high-dimensional latent space to approximate the algorithm.
> >   - This allows our model to outperform the approximation algorithm even generalized to larger instances.
> >   - It also allows the model embedded with algorithmic knowledge to be applied to real-world problems with high-dimensional inputs, bypassing feature engineering of traditional algorithms or dimension reduction of hardcoding it. It also emphasizes our motivation to extend NAR to NP-hard, where most real-world problems fall under.
> >
> > Thank you once again for your positive feedback on our methodology and writing. We deeply appreciate your constructive feedback which greatly strengthens our paper. We also revised our manuscript in Introduction, Related work, Appendix H & I to incorporate the changes. Please let us know if you have any other questions and thanks for your time!

---

> ### Author Response · Authors · 2024-11-30
>
> Dear Reveiwer EkCx, we are writing to gently ask the reviewer if we have addressed your questions and concerns. If so, we wonder if the reviewer could kindly reconsider the rating based on our rebuttal and General Response. Once again, we are deeply grateful to the reviewer for their time spent and their invaluable feedback which greatly strengthened our paper!

---

### Official Review · Reviewer_KsRr · 2024-10-26

**Soundness:** 3
**Presentation:** 3
**Contribution:** 3
**Rating:** 6
**Confidence:** 2

**Summary:**

This paper design a GNN model based on the primal-dual algorithm to solve CO problems including Minimum Vertex Cover, Minimum Set Cover, and Minimum hitting set. The retrained model can be used to warm start a commercial solvers, and also applicable to other problems  closely connected with the studied problems, eg node classification.

**Strengths:**

Writing is good with many details. The method looks sound. Experiments are comprehensive.

**Weaknesses:**

- To me, it is not clear whether the work is important in literature, especially in dealing with the three problems.

- Important baselines are missing. For example, other neural solvers for the three problems are not provided. GIN is not designed with the purpose of solving the problem. So the comparison with merely GIN can be unfair. To demonstrate the effectiveness, it is necessary to include more solvers for the problem.

- Authors give results of model performance on OOD data in Table 2. However, since authors do not give results of other baselines, the OOD generalizability cannot be proved directly. By the way, no OOD results on the minimum set cover problem are provided.

**Questions:**

- The three problems are closely related. Can the method be applied on other problems that can work with the  primal-dual algorithm?

- In table 2, why use geometric mean for total time while arithmetic mean for optimal time?

- what is the relation of PDGNN and GAT, GCN and SAGE? How are they combined as shown in Table 4? What are the inputs and outputs of each module? Since the main text does not mention how are PDGNN works with other models, the results confused me.

- what does the claimed OOD generalizability comes from? There is no module designed for the purpose. So it confuses me why the method has good OOD generalizability.

Since I am not an expert in these problems, I would consider raising my score based on the rebuttal.

---

> ### Author Response · Authors · 2024-11-21
> **Rebuttal (Part 1)**
>
> We thank Reviewer KsRr for their kind words regarding the quality and detail of our writing, the soundness of our method, and the comprehensiveness of our experiments. We also appreciate the thoughtful concerns and suggestions from Reviewer KsRr, which we have carefully considered while conducting additional experiments. Revisions have been made to the manuscript, with changes highlighted in blue. Below, we respond to the raised concerns point by point, offering further context and a detailed discussion of our model's contributions and evaluation.
>
> **1. Importance in literature:**
>
> We appreciate Reviewer KsRr for raising this important question, which allows us to clarify our work's primary focus and contributions. Our research centers on Neural Algorithmic Reasoning (NAR) — designing novel architectures to simulate algorithms and investigating the algorithmic reasoning capabilities of GNNs. As discussed in detail in **General Response (Point 1)**, prior work in NAR has predominantly focused on simulating algorithms for polynomial-time-solvable problems, such as the Bellman-Ford algorithm for the shortest-path problem. In contrast, we propose a general NAR framework that extends supervised NAR into the domain of NP-hard problems. This extension represents a significant step forward because the primal-dual approximation algorithm we employ is not limited to a specific problem — it generalizes to many traditional algorithms for both polynomial-time-solvable and NP-hard problems. Thus, our work is not primarily about solving three CO problems; rather, it is about broadening the scope of NAR to tackle a previously underexplored domain. We believe this advancement opens up exciting new directions for applying NAR to more complex problem spaces, making a substantial contribution to the field.
>
> We also recognize that the NP-hard problems we study overlap with Neural Combinatorial Optimization (NCO), which focuses on learning task-specific heuristics or end-to-end optimization. While NCO is not the primary focus of our work and remains fundamentally distinct from NAR, we discuss how our method contributes to NCO in **General Response (Point 2)**.
>
> We are grateful to Reviewer KsRr for highlighting this concern. In response, we have revised the Introduction and Related Works sections to elaborate on our motivation rooted in NAR and to clarify how our contributions relate to and differ from NCO. By doing so, we hope to provide a clearer understanding of the unique value our work brings to both fields.
>
> **2. Comparison with other baselines:**
>
> We value Reviewer KsRr’s concern regarding additional baselines. We want to clarify that the baseline selections for Table 1 align with standard practices in NAR literature, as explained in General Response (Point 1). Furthermore, to directly address Reviewer KsRr’s concern, we have conducted additional experiments on the RB benchmark graphs (well-known hard instances for MVC), and compared our method with several NCO baselines. We have included the results in the **General Response (Point 3)**. We appreciate Reviewer KsRr for raising this point, as it provides an opportunity to clarify the validity of our evaluation settings and further emphasize the strengths of our method.

---

> ### Author Response · Authors · 2024-11-21
> **Rebuttal (Part 2)**
>
> **3. OOD generalization:**
>
> We thank Reviewer KsRr for raising this important clarification question. In the NAR literature, OOD generalization is typically defined along two dimensions: size generalization and graph distribution generalization. Our work addresses both aspects within the scope of our experiments (Table 1 and Table 2). The OOD generalization is reflected by (i) comparing with No Algo baselines, we show algorithmic knowledge strengthens generalization (ii) comparing with the approximation algorithm baseline, our model consistently outperforms it when generalized to larger ones and other graph distributions. For the cases of MSC and MHS, synthetic graphs were generated using bipartite graphs for training. Unlike MVC, where graph distribution generalization is well-defined (e.g., training on Barabasi-Albert graphs and generalizing to Erdos-Renyi graphs), there are fewer standard baselines for generating out-of-distribution bipartite graphs. Despite this, we demonstrated size generalization in Table 1 and further validated our method through additional ablations on larger graph sizes in Table R2. We appreciate the reviewer for highlighting this point, as it allows us to clarify the scope of our contributions and the methodology employed in our work.
>
> Table R2: Total weight of PDGNN / total weight of primal-dual algorithm (lower the better).
>
> |               | 128 nodes      | 256 nodes      | 512 nodes      | 1024 nodes     |
> |---------------|----------------|----------------|----------------|----------------|
> | MVC           | 0.973 ± 0.004 | 0.971 ± 0.005 | 0.969 ± 0.005 | 0.971 ± 0.005 |
> | MSC           | 1.001 ± 0.012 | 0.999 ± 0.008 | 0.996 ± 0.007 | 0.998 ± 0.009 |
> | MHS           | 0.984 ± 0.004 | 0.982 ± 0.004 | 0.987 ± 0.005 | 0.988 ± 0.005 |
>
> **Answers to questions:**
> - Q1: Yes, our method proposes a general framework can be applied to a broad range of problems that can be tackled by the primal-dual approximation algorithms. This is not limited to NP-hard problems, but also includes polynomial-time-solvable problems. For example, Kruskal’s algorithm for Minimum Spanning Tree is an instantiation of the primal-dual algorithm with the uniform increase rule. This highlights our innovation of the uniform increase rule, which allows our model to cover an even broader range of algorithms with the framework. We also included a more in-depth discussion on the classes of problems that our framework can be applied to in Appendix F.
> - Q2: We follow the two standard metrics used in the literature. The geometric mean [1] is commonly used because it is less sensitive to outliers, for example, when the time limit is hit and optimal solutions are not found. Arithmetic means are used [2] to show a more straightforward representation of runtime when optimal solutions are found. We included both metrics to provide a more comprehensive analysis to demonstrate the runtime benefit.
> - Q3: We apply GAT, GCN, and SAGE (base models) in the standard way for the three node-classification tasks, where the inputs are node features and the outputs are predicted classes. Recall that NAR follows an encoder-(GNN) processor-decoder framework. For PDGNN, it is pretrained on synthetic algorithmic data to simulate the primal-dual algorithm. After pretraining, the GNN processor is frozen, the encoder is replaced with a new one, and the decoder is discarded. The new encoder takes node features as input, maps them to the latent space where the GNN processor simulates the primal-dual algorithm, and produces node embeddings that encode algorithmic knowledge. These embeddings are concatenated with the embeddings generated by GAT/GCN/SAGE, and the combined embeddings are passed through a linear layer to predict classes. This framework is based on [3][4], and we have improved Section 5.4 to clarify how PDGNN integrates with the base models.
> - Q4: Further elaborating on Point 3, one key motivation of NAR is to demonstrate how embedding algorithmic knowledge can enhance OOD generalization. An analogy can be drawn to how humans learn the underlying principles of algorithms to solve a wide range of problems and apply this understanding to new problem instances. Similarly, we show in Table 1 that PDGNN, when trained with algorithmic supervision, exhibits significantly better generalization compared to the "No Algo" baseline.

---

> > ### Author Response · Authors · 2024-11-21
> > **Rebuttal (Part 3)**
> >
> > We sincerely thank Reviewer KsRr for their positive feedback on the quality and detail of our writing, the soundness of our methodology, and the thoroughness of our experiments. We also greatly appreciate the constructive suggestions and thoughtful clarification questions, which have been invaluable in improving both our experiments and the presentation of our contributions. We hope that the additional experiments and explanations address your concerns and offer greater clarity on the strengths of our work. We look forward to your feedback and remain happy to address any further questions to ensure our contributions are fully understood and effectively communicated!
> >
> > [1] Gasse et al., Exact combinatorial optimization with graph convolutional neural networks, NeurIPS 2019.
> >
> > [2] Huang et al., Distributional miplib: a multi-domain library for advancing ml-guided milp methods, 2024.
> >
> > [3] Veličković and Blundell, Neural Algorithmic Reasoning, Patterns, 2021.
> >
> > [4] Numeroso et al., Dual Algorithmic Reasoning, ICLR 2023.

---

> > > ### Author Response · Authors · 2024-11-25
> > >
> > > Dear Reviewer KsRr,
> > >
> > > Hope this note finds you well! We're just checking in since it is almost the end of the rebuttal period and we were wondering if we have addressed your main concerns.
> > >
> > > To recap, we believe the main concerns are:
> > > - **Importance of our work in literature**: We elaborated on our motivation by providing more context on NAR. Our method is not constrained by the three problems, but a general framework that extends NAR to a previously underexplored domain, largely expanding its applicability to real-world problems which are commonly NP-hard in nature.
> > > - **Comparison with other baselines**: We explained that our baseline choices align with previous NAR literature. However, we also conducted additional experiments on hard instances for MVC and compared with several NCO baselines.
> > > - **OOD generalization**: We explained that OOD generalization in NAR literature is commonly defined in terms of size and graph distribution generalization, which we both evaluated in experiments. We also provided additional ablation by scaling up the size of the graphs to demonstrate our generalization advantage.
> > >
> > > Thank you once again for your positive feedback on our writing, methodology and experiments. We deeply appreciate your constructive feedback which greatly strengthens our paper. We also revised our manuscript in Introduction, Related work, Appendix H & I to incorporate the changes. Please let us know if you have any other questions and thanks for your time!

---

> > > > ### Comment · Reviewer_KsRr · 2024-11-25
> > > > **Comment by reviewer KsRr**
> > > >
> > > > Thanks for the detailed response. I fully believe in the author's experimental results regarding the model's generalization ability, but to me the question is **"how"** and **"why"**. Hope the authors can give more theoretical analysis or hints on the point.

---

> ### Author Response · Authors · 2024-11-26
> **Reply to additional question**
>
> We thank the reviewer for the important question on understanding the "how" and "why" behind our model's generalization ability.
>
> **Algorithmic alignment**: Our method is grounded on the theoretical insights from [5], where they proposed a formal theoretical framework of algorithmic alignment (Definition 3.4 of [5]) between the network and algorithmic structure. In Theorem 3.6 of [5], they showed that better algorithmic alignment leads to better generalization and sample efficiency (the network only needs to learn a single reasoning step to simulate the entire algorithm). As an example, in Figure 2 of [5], they used the alignment between GNN and Bellman-Ford algorithm to explain why GNN generalizes when learning to answer shortest paths.
>
> **GNN extrapolation**: Furthermore, in [6], they suggest a hypothesis with theoretical and empirical evidence that, GNNs can extrapolate well in algorithmic tasks to new data (e.g. larger graphs) if we encode appropriate non-linearities in the architecture and input representation (through domain knowledge or representation learning), so that the MLP modules in GNNs only need to learn nearly linear transformations. An illustrative example can be found in Figure 2 of [6].
>
> In our work, we design the architecture to align well with the primal-dual algorithm. We show in Theorem 1 that each step of the primal-dual algorithm can be decomposed into eight message-passing layers. In this decomposition, all non-linearities are embedded within the model architecture and input representations, leaving the learnable components as purely linear layers. Our architecture aligns with the theoretical findings of [5][6], as it strengthens algorithmic alignment and simplifies the learning task for the MLP modules to linear operations. Thus, our paper follows [5]’s algorithmic alignment framework and provides additional validation of [6]'s hypothesis, offering a deeper theoretical explanation for why our model extrapolates effectively. These provide theoretical insights in explaining our model’s generalization beyond the training data, which is reflected in our experimental results.
>
> We hope this answers the reviewer’s question. We will include these explanations in Appendix G, where we discuss related works on NAR. Please let us know if you have any other questions and thanks for your time!
>
> [5] Xu et al., What can neural networks reason about? ICLR 2020.
>
> [6] Xu el al., How neural networks extrapolate: from feedforward to graph neural networks, ICLR 2021.

---

> ### Author Response · Authors · 2024-11-30
>
> Dear Reveiwer KsRr, we are writing to gently ask the reviewer if we have addressed your questions and concerns. If so, we wonder if the reviewer could kindly reconsider the rating based on our rebuttal and General Response. Once again, we are deeply grateful to the reviewer for their time spent and their invaluable feedback which greatly strengthened our paper!

---

### Official Review · Reviewer_YZYW · 2024-11-01

**Soundness:** 3
**Presentation:** 3
**Contribution:** 3
**Rating:** 5
**Confidence:** 4

**Summary:**

This paper proposes an approach that integrates the primal-dual method with GNNs to solve NP-hard combinatorial optimization problems more efficiently.
It establishes theoretical guarantees that the proposed framework can replicate the classical primal-dual method.
By refining the optimal solutions for these problems, the model surpasses the performance of conventional algorithms.
Additionally, by utilizing predictions as warm starts, the search time for commercial solvers can be reduced.
Numerical results showcase the scalability and effectiveness of the proposed framework.

**Strengths:**

1. The paper proposes an approach that is the first NAR method to surpass the learned algorithm.
2. Author conducted several experiments with good numerical results favoring the proposed framework.
3. Theoretical proofs are given to validate the effectiveness of the proposed framework.

**Weaknesses:**

While the proposed framework is designed to address NP-hard combinatorial optimization problems, the proposed framework primarily operates on linear programming (LP) relaxation, which appears to be an extension to existing methods, such as those referenced in [1] and [2].
Besides, there are also many works that directly predicts solutions to CO problems. e.g. [3] and [4].
An extended discussion, especially with experiments, on how the proposed approach differs from them would strengthen the authors' claims.

[1]. Bingheng Li, Linxin Yang, Yupeng Chen, Senmiao Wang, Qian Chen, Haitao Mao, Yao Ma, Akang Wang, Tian Ding, Jiliang Tang, and Ruoyu Sun. Pdhg-unrolled learning-to-optimize method for large-scale linear programming, 2024.

[2]. Ziang Chen, Jialin Liu, Xinshang Wang, Jianfeng Lu, and Wotao Yin. On representing linear programs by graph neural networks, 2023.

[3].  Vinod Nair, Sergey Bartunov, Felix Gimeno, Ingrid von Glehn, Pawel Lichocki, Ivan Lobov, Brendan O’Donoghue, Nicolas Sonnerat, Christian Tjandraatmadja, Pengming Wang, Ravichandra Addanki, Tharindi Hapuarachchi, Thomas Keck, James Keeling, Pushmeet Kohli, Ira Ktena, Yujia Li, Oriol Vinyals, and Yori Zwols. Solving mixed integer programs using neural networks, 2021.

[4]. Dinghuai Zhang, Hanjun Dai, Nikolay Malkin, Aaron Courville, Yoshua Bengio, and Ling Pan. Let the flows tell: Solving graph combinatorial optimization problems with gflownets, 2023.

**Questions:**

Please refer to "Weakness".

---

> ### Author Response · Authors · 2024-11-21
> **Rebuttal (Part 1)**
>
> We would like to thank Reviewer YZYW for highlighting the effectiveness of our framework and acknowledging the favorable numerical results from extensive experiments and theoretical proofs. We are also grateful to Reviewer YZYW for their thoughtful suggestions, which we have thoroughly considered and addressed by conducting additional experiments. We have revised our manuscript with the changes highlighted in blue. Below, we provide detailed responses to the concerns raised, offering further context and an in-depth discussion of our model's contributions and evaluation.
>
>
> **1. Difference with LP literature:**
>
> We appreciate Reviewer YZYW’s comments and the reference to works on LP relaxations. While we acknowledge the connection between CO problems and LP, as many CO problems can be represented using LP formulations, and primal-dual algorithms build on these relaxations, our focus and methodology is distinct. Our primary goal is on Neural Algorithmic Reasoning (NAR), by embedding algorithmic knowledge into GNNs and analyzing their algorithmic reasoning ability. For more context on NAR, we refer Reviewer YZYW to **General Response (Point 1)**. This focus distinguishes our work from approaches designed specifically to optimize LPs. We have also added a paragraph in Appendix I to discuss LP-related works and clarify how our approach differs.
>
> More specifically, [1] introduces a Learning-to-Optimize method to mimic first-order methods for solving large-scale LPs. While [1] focuses on developing efficient solvers for LPs, we aim to simulate the primal-dual approximation algorithm for NP-hard problems using GNNs. Although both approaches reference the primal-dual terminology, this similarity is not straightforward. Duality is widely used in optimization, but our work applies it to study algorithmic reasoning. For example, the primal-dual approximation algorithm can be instantiated to many traditional algorithms, such as Kruskal’s algorithm for MST. Furthermore, unlike [1], our method relies on intermediate supervision from the primal-dual algorithm to guide reasoning, ensuring that the model learns to mimic algorithmic steps. Additionally, we incorporate optimal solutions into the training process to improve solution quality. Moreover, the architectures differ significantly: our method employs a recurrent application of a GNN to iteratively solve problems, while [1] does not use GNNs or recurrent modeling. These distinctions highlight that our focus is not on solving LPs but on leveraging NAR to generalize algorithmic reasoning for NP-hard problems.
>
> Similarly, while [2] explores using GNNs to represent LPs, their work explicitly avoids integer programs and does not address NP-hard problems. In contrast, our work studies integer programming, where solutions are inherently more complex and computationally challenging. While [2] emphasizes the power of GNNs to represent LPs, our approach embeds the primal-dual approximation algorithm into a GNN framework to tackle NP-hard problems, evaluate generalization, and study the robustness of algorithmic reasoning. These differences in the overall goal and methodology make our work distinct from [2].

---

> ### Author Response · Authors · 2024-11-21
> **Rebuttal (Part 2)**
>
> **2. Comparison with NCO baselines:**
>
> We value Reviewer YZYW’s feedback on comparison with Neural Combinatorial Optimization (NCO) baselines, which directly predict solutions to CO problems. However, we emphasize our primary focus remains on Neural Algorithmic Reasoning (NAR), and our baseline selection closely follows prior NAR works. While not our primary focus, we positioned our method within the context of NCO in **General Response (Point 2)**, highlighting our contribution to an under-explored area within NCO. We have included a discussion of our work on NCO in Related Works and Appendix H in our revised manuscript.
>
> To directly address Reviewer YZYW’s concern, we also have conducted additional experiments on the RB benchmark graphs (well-known hard instances for MVC), and compared our method with several NCO baselines. We have included the results in **General Response (Point 3)**. We thank Reviewer YZYW for the suggestion, which further emphasizes the strengths of our approach on data efficiency and strong generalization capabilities.
>
> We also want to clarify why [3] and [4] were not chosen as baselines. [3] explores replacing aspects of branch-and-bound with a GNN, which is not a polynomial-time approach, whereas our objective is to learn polynomial-time method using a GNN by simulating a polynomial-time approximation algorithm. Regarding [4], since our method falls under the category of GNN-based supervised learning for NCO, GFlowNets is not a GNN architecture. As explained in General Response (Point 2), our approach addresses an underexplored area in NCO by leveraging GNNs with intermediate supervision from an algorithm rather than end-to-end learning, effectively tackling the challenges of data efficiency and generalization.
>
> We sincerely thank Reviewer YZYW for their positive acknowledgment of our methodology, experimental results, and theoretical proofs. We also appreciate the constructive suggestions, which have significantly helped us enhance both our experiments and the presentation of our contributions, particularly by contrasting them with related works. We hope that the additional experiments and clarifications provided address your concerns and offer a clearer perspective on the strengths of our work. We look forward to hearing your feedback soon and are happy to address any further questions you may have to ensure our work is fully understood and its contributions are clearly conveyed!
>
> [1]. Li & Yang et al., PDHG-Unrolled Learning-To-Optimize Method For Large-Scale Linear Programming, ICML 2024.
>
> [2]. Chen, et al., On Representing Linear Programs By Graph Neural Networks, 2023.
>
> [3]. Nair et al., Solving Mixed Integer Programs Using Neural Networks, 2021.
>
> [4]. Zhang et al., Let The Flows Tell: Solving Graph Combinatorial Optimization Problems with GFlowNets, NeurIPS 2023.

---

> > ### Comment · Reviewer_YZYW · 2024-11-23
> >
> > Thank you for the clarification. However, I remain unconvinced that [3] is entirely irrelevant. The neural diving approach in [3] employs an estimation heuristic for obtaining primal solutions to general MILPs, which covers problems discussed in this work. Moreover, the additional results indicate that PDGNN does not outperform Gurobi, even with its 2-second time limit. This raises additional concerns about the effectiveness of PDGNN. Given that primal-dual methods are particularly efficient for large-scale problems, a demonstration on such cases might enhance the strength and applicability of this work.

---

> ### Author Response · Authors · 2024-11-23
> **Reply to additional questions**
>
> Thank you for the follow-up questions! We appreciate the opportunity to clarify. While we agree that [3] overlaps with the NP-hard problems addressed in our work, our goal is different, as we will explain below. Additionally, commercial solvers such as Gurobi are known for their efficiency and are challenging to surpass, particularly for fully ML-based approaches [5]. Therefore, we believe it is noteworthy that our method achieves results catching up with Gurobi, especially given that each problem instance is computed in ~ 0.02 sec.
>
> 1.We would like to respectfully emphasize that our primary objective is *neither* to propose a more efficient solver for optimization problems, *nor* to focus on ML-facilitated MILP approaches. Instead, our main focus is to study the algorithmic reasoning capabilities of neural networks (General Response Point 1 & 2). On a higher level of difference, MILP focuses on finding exact or approximate solutions to a well-defined mathematical formulation. In contrast, our work concerns a machine learning problem with focus on training with algorithmic data and aims to generalize from observed data to unseen examples. For a related example, **we recommend [7]**, which also uses the duality principle from LP to facilitate NAR.
>
> 2.Furthermore, there are several advantages of NAR compared with traditional algorithms and solvers.
> - Motivation: As described in Section 5.4, a key motivation for NAR is to empower models with algorithmic reasoning capabilities, enabling them to address real-world tasks that require structured reasoning.
>   - The NAR framework comprises an encoder, a GNN-based processor, and a decoder. During pretraining, the encoder maps inputs (e.g., weights) into a high-dimensional latent space, where the GNN simulates the algorithm operation (e.g. primal-dual algorithm). The decoder maps this output back to produce solutions. This pretrained GNN, imbued with algorithmic knowledge, can then be applied to real-world datasets by replacing the encoder and decoder with components that map the new input features to and from the shared high-dimensional latent space. For an illustrative example, please refer to **Figure 1 of [6]**.
>
> - Advantages of NAR over algorithms and solvers:
>
>   - *Direct operation on raw input features:* Traditional algorithms or solvers often require feature engineering given raw input features or formulation as an optimization problem. In contrast, by simulating the algorithmic behavior on a high-dimensional space, NAR can directly operate on the raw input features, handling noise or incomplete data, and accounting for uncertainty or patterns in real-world data.
>   - *Flexibility beyond optimization objectives:* Algorithms and solvers are typically applied to optimization tasks involving minimization or maximization objectives. NAR, however, embeds algorithmic knowledge in a way that can be extended to other tasks. For example, in our case, the pretrained PDGNN is used to produce node embeddings for a node classification task. Here, vertex-cover-like information enhances prediction, even though the task itself is not an optimization problem.
>
> - Therefore, NAR has the advantage to combine the benefits from flexibility of deep learning and structured reasoning of traditional algorithms. It also further motivates our extension of NAR to the NP-hard domain, given the prevalence of NP-hard problems in real-world applications.
>
> Thank you once again for raising these important points. In our revised manuscript, we edited Introduction, Related Works, Appendix H & I to reflect these changes and clarify our goals. We hope this response clarifies the distinctions between our goals and traditional solver-based approaches while emphasizing the unique advantages of NAR. Please share with us any further questions or concerns—we are happy to address them.
>
> [5] Wang & Li, Unsupervised Learning for Combinatorial Optimization Needs Meta Learning, ICLR 2023.
>
> [6] Veličković and Blundell, Neural Algorithmic Reasoning, Patterns, 2021.
>
> [7] Numeroso et al., Dual Algorithmic Reasoning, ICLR 2023.

---

> > ### Author Response · Authors · 2024-11-26
> >
> > Dear Reviewer YZYW,
> >
> > Hope this note finds you well! We're just checking in since it is almost the end of the rebuttal period and we were wondering if we have addressed your main concerns.
> >
> > To recap, we believe the main concerns are:
> > - **Focus of our paper**: We clarified that our primary focus is on NAR. Our focus is neither to propose a more efficient solver for optimization problems, nor to focus on ML-facilitated MILP approaches, but to study the algorithmic reasoning capabilities of neural networks. Our baseline selection and experiments follow previous NAR literature, e.g. [7].
> > - **Comparison with additional baselines**: Though not our main focus, we also explained how our method potentially contributes to NCO. We also conducted additional experiments on hard instances for MVC and compared with several NCO baselines.
> > - **Advantages of NAR**: We elaborated the distinct advantages of NAR, especially how pretraining an NAR model with algorithmic knowledge can directly apply it to real-world datasets that are not necessarily an optimization problem. This helps to bypass the need for feature engineering or mathematical formulation required by traditional algorithms and solvers. This highlights the importance of our work - extending NAR to an underexplored domain and largely expanding NAR’s applicability because many real-world problems are NP-hard in nature.
> >
> > Thank you once again for your positive feedback on our methodology, experimental results, and theoretical proofs. We deeply appreciate your constructive feedback which greatly strengthens our paper. We also revised our manuscript in Introduction, Related work, Appendix H & I to incorporate the changes. Please let us know if you have any other questions and thanks for your time!

---

> ### Author Response · Authors · 2024-11-30
>
> Dear Reveiwer YZYW, we are writing to gently ask the reviewer if we have addressed your questions and concerns. If so, we wonder if the reviewer could kindly reconsider the rating based on our rebuttal and General Response. Once again, we are deeply grateful to the reviewer for their time spent and their invaluable feedback which greatly strengthened our paper!

---

### Official Review · Reviewer_Y8qc · 2024-11-04

**Soundness:** 2
**Presentation:** 2
**Contribution:** 1
**Rating:** 5
**Confidence:** 4

**Summary:**

The paper proposes to simulate the primal-dual framework for approximation algorithms by Goemans and Williamson using a graph neural network. The paper follows the standard neural algorithmic reasoning blueprint with an encoder-processor-decoder approach and it leverages a gnn that performs message passing on a bipartite graph representation of the problem. The model is used to solve NP-hard problems such as minimum vertex cover, minimum hitting set, and minimum set cover. The model also leverages optimal solutions to the problem as additional supervision and is tested on real and synthetic graphs.

**Strengths:**

- If I understand correctly, this primal dual approach will necessarily yield a certificate of solution quality since the dual objective bounds the primal, which is not easy to obtain with neural nets on those problems.
- Ablations and experiments on real world data are helpful.

**Weaknesses:**

My main concerns are the following:
- The experimental comparisons are inadequate. For example, MVC (and its complement problem of max independent set) have been tackled by several works in the literature (examples can be found in [1,2,3]). Warm-starting solvers has also been done (e.g., see 4). In my view,  discussion and comparisons with some work from the literature on the core experiments as well as the warm start ones would help clarify where the model stands overall. Furthermore, the experiments on synthetic instances are rather small-scale (especially table 1). [2,3] provide an experimental setting with hard synthetic instances of larger sizes for MVC. It would be good to see how the model performs against some of those baselines on those instances.
- On the topic of scale, it would be good to see what the performance looks like if the synthetic instances were scaled up an order of magnitude. Right now, it seems like it would take a lot of supervision to train a scalable version of this algorithm, especially if one wanted to leverage optimal solutions, which become costly for NP-hard problems as the instance size grows.
- A central goal of this paper is to have a machine-learning model that closely aligns with the primal-dual framework. The numbers seem to agree in table 1 when it comes to MVC and MHS but for MSC we see that not having access to the optimal solutions seems to lead to results worse than the primal-dual algorithm itself. This points to the broader issue that even though the model is designed to closely align with this framework, the approximation guarantee is not secured in practice.

Overall, this is an interesting approach to neural combinatorial optimization. However, I find the empirical results unconvincing at this stage and ultimately the algorithm does not seem to preserve the approximation guarantee in practice so I am unsure about the contribution as a whole. At this stage. I cannot recommend acceptance but I am open to reconsidering after the rebuttal.




1. Khalil, Elias, et al. "Learning combinatorial optimization algorithms over graphs." Advances in neural information processing systems 30 (2017).
2. Brusca, Lorenzo, et al. "Maximum independent set: self-training through dynamic programming." Advances in Neural Information Processing Systems 36 (2023)
3. Yau, Morris, et al. "Are Graph Neural Networks Optimal Approximation Algorithms?." arXiv preprint arXiv:2310.00526 (2023).
4. Benidis, Konstantinos, et al. "Solving recurrent MIPs with semi-supervised graph neural networks." arXiv preprint arXiv:2302.11992 (2023).

**Questions:**

Do the mean-max aggregation lines in table 1 use optimal solutions in training as well? I find it a little strange that max aggregation performs better than the algorithm without the optimal solutions. Shouldn't the problem-specific aggregations do better?

---

> ### Author Response · Authors · 2024-11-21
> **Rebuttal (Part 1)**
>
> We are grateful to Reviewer Y8qc for recognizing the effectiveness and novelty of our framework and finding our real-world experiments helpful. We also appreciate the detailed suggestions provided by Reviewer Y8qc, which we have taken into careful consideration and conducted additional experiments. We also edited our manuscript and highlighted the changes in blue. Below, we address the concerns raised point by point, providing additional context and discussing our model's contributions and evaluation in more detail.
>
> **1. Position of our work in literature:**
>
> We sincerely appreciate Reviewer Y8qc for acknowledging our method as “an interesting approach to neural combinatorial optimization”. However, there seemed to be a misunderstanding: our primary focus is Neural Algorithmic Reasoning (NAR) rather than Neural Combinatorial Optimization (NCO). We would like to direct Reviewer Y8qc to our **General Response (Point 1)**, where we provided more background on NAR and highlighted our contribution to NAR. We thank Reviewer Y8qc for raising the concern, and we have significantly edited the Introduction and Related Works to emphasize our motivation and position our method better in the context of NAR.
>
> We also understand that there is an overlap in the problems we study with NCO. Therefore, though not our main focus, we motivate our approach as a data-efficient and robust supervised learning framework for NCO in **General Response (Point 2)**. Again, we thank Reviewer Y8qc for acknowledging our contribution to NCO, which broadens the potential impact of our method. We also included a discussion of our work to NCO in Related Works and Appendix H to reflect the changes.
>
> **2. Hard instances for MVC and NCO baselines:**
>
> We are grateful to Reviewer Y8qc for suggesting hard MVC instances. Based on Reviewer Y8qc’s suggestion, we have conducted additional experiments on the RB benchmark graphs, and compared our method with several NCO baselines. We have included the results in the **General Response (Point 3)**. We thank Reviewer Y8qc for pointing us to relevant benchmarks and baselines, which further emphasize the data efficiency and strong generalization capabilities of our approach.
>
> **3. Scalability:**
>
> We thank Reviewer Y8qc for raising scalability concerns as it highlights the strength of our method: data efficiency and robustness from algorithmic reasoning. We conducted additional experiments by increasing the test graph sizes to 1024 while keeping the training set limited to graphs of size 16. Due to the large sizes, the optimal solutions are unavailable due to computational limitations for comparison (however, we do provide results comparing our model to optimal solutions on hard instances of large RB graphs for MVC). Despite this significant increase in scale, our model consistently performed well, demonstrating strong size generalization based on its algorithmic knowledge. This underscores the data efficiency of our method, where supervision signals were effectively derived from a polynomial-time approximation algorithm and optimal solutions from small problem instances.
>
> Table R2: Total weight of PDGNN / total weight of primal-dual algorithm (lower the better).
>
> |               | 128 nodes      | 256 nodes      | 512 nodes      | 1024 nodes     |
> |---------------|----------------|----------------|----------------|----------------|
> | MVC           | 0.973 ± 0.004 | 0.971 ± 0.005 | 0.969 ± 0.005 | 0.971 ± 0.005 |
> | MSC           | 1.001 ± 0.012 | 0.999 ± 0.008 | 0.996 ± 0.007 | 0.998 ± 0.009 |
> | MHS           | 0.984 ± 0.004 | 0.982 ± 0.004 | 0.987 ± 0.005 | 0.988 ± 0.005 |
>
> Table R3: Computation time (seconds per graph) of generating solutions with primal-dual algorithm (Algo) and PDGNN (Model).
>
> |               | 128  |          | 256  |           | 512  |       | 1024  |           |
> |---------------|-----------|------------|-----------|------------|-----------|------------|-----------|------------|
> |               | Algo      | Model      | Algo      | Model      | Algo      | Model      | Algo      | Model      |
> | MVC           | 0.12      | 0.01       | 0.15      | 0.02       | 0.20      | 0.02       | 0.44      | 0.03       |
> | MSC           | 0.10      | 0.01       | 0.16      | 0.02       | 0.27      | 0.02       | 0.45      | 0.02       |
> | MHS           | 0.15      | 0.01       | 0.24      | 0.02       | 0.32      | 0.02       | 0.52      | 0.03       |

---

> > ### Author Response · Authors · 2024-11-21
> > **Rebuttal (Part 2)**
> >
> > **4. Approximation guarantee:**
> >
> > We agree with Reviewer Y8qc that our method may not preserve the worst-case approximation guarantees of the primal-dual algorithm in practice, as pointed out in the footnote of Page 2. Instead, our model focuses on learning solutions that perform better for the training distribution and generalize well to new instances. While this does not mean that the worst-case guarantees are secured, it is important to note that such cases are often less common in real-world scenarios. One of the core strengths of NAR lies in leveraging a pretrained GNN with embedded algorithmic knowledge to tackle standard node-classification tasks on real-world datasets (Section 5.4). This highlights the value of training models to produce high-quality solutions for non-worst-case instances, even if they do not preserve worst-case guarantees. This aligns with the overarching goals of NAR. We thank the reviewer for raising the question, and we have included a discussion on approximation guarantees in Limitation (Appendix F).
> >
> > **Answers to question:**
> > - Q: Do the mean-max aggregation lines in table 1 use optimal solutions in training as well?
> > - A: Yes, the mean-max aggregation lines in Table 1 also incorporate optimal solutions. As noted in [1], max aggregation is particularly effective for tasks that require identifying representative nodes or the “skeleton”, which may explain its superior performance compared to PDGNN (no optm) for MVC, as it better captures in-distribution optimal solutions (16 nodes). However, we note that PDGNN with algorithmic knowledge demonstrates stronger generalization than max aggregation, emphasizing the importance of algorithmic alignment in enhancing generalization.
> >
> > We are grateful to Reviewer Y8qc for positively acknowledging our method as interesting. We are also thankful for the constructive suggestions and resources that allowed us to greatly strengthen both our experiments and the presentation of our contributions. We deeply value Reviewer Y8qc’s concerns, and we hope our additional experiments and clarifications address your questions and provide a clearer understanding of the strengths of our work. We look forward to your insights and are ready to address any further points you may have to ensure our contributions are effectively conveyed!
> >
> > [1] Xu & Hu et al., How Powerful Are Graph Neural Networks? ICLR 2019.

---

> > > ### Author Response · Authors · 2024-11-25
> > >
> > > Dear Reviewer Y8qc,
> > >
> > > Hope this note finds you well! We're just checking in since it is almost the end of the rebuttal period and we were wondering if we have addressed your main concerns.
> > >
> > > To recap, we believe the main concerns are:
> > > - **Position of our work on NAR and NCO**: We clarified that our primary focus is on NAR, and made its distinction with NCO. We also elaborated how our method can potentially contribute to NCO.
> > > - **Hard instances and additional baselines**: We conducted additional experiments on hard instances for MVC following your suggestion of RB graphs, and compared with several NCO baselines.
> > > - **Scalability of our method**: We conducted additional experiments and showed our method can scale to graphs of 1024 nodes when trained on graphs of 16 nodes for all three problems.
> > > - **Approximation ratio preservation**: We explained that though our model may not preserve the worst-case approximation guarantee in practice, our focus is on learning solutions that perform better for the training distribution and generalize well to new instances. This aligns with NAR’s motivation to install algorithmic knowledge to real-world datasets where worst-case scenarios are less common.
> > >
> > > Thank you once again for recognizing our method as “interesting” and providing constructive feedback. We found your suggestions instrumental in strengthening our paper. We also revised our manuscript in Introduction, Related work, Appendix H & I to incorporate the changes. Please let us know if you have any other questions and thanks for your time!

---

> ### Comment · Reviewer_Y8qc · 2024-11-25
>
> I thank the authors for their detailed response and for engaging with my review. Let me explain my rationale for how I'm assessing the paper. The paper learns to essentially emulate a classical primal-dual algorithm that is known to have an approximation guarantee. My goal is to understand what is the main contribution of such work. With that in mind, there are a few axes along which I'm judging the paper:
>
> 1. How does it perform against the actual algorithm (both theory and practice)?
> 2. What does this buy us over running the algorithm?
> 3. How does the method compare in the broader context of neural network methods that solve such problems?
>
>
> The evaluation wrt point 3 was found lacking (although it has improved post-rebuttal).
> The authors argue that perhaps we shouldn't put all our focus on point 3 because there are aspects of the work (such as 1,2) that might be worthwhile.  This is more generally a common theme for NAR and perhaps it is fair to some extent.
>
> But that's my main concern.  Part of the reason I think it's important to focus on point 3 is because I need a more diverse evaluation of out of distribution performance. By OOD I don't just mean size generalization. Even if we are only to judge how well the proposed model learns an algorithm, then surely looking at evaluations on realistic test instances should give us useful information. The OOD experiments for other graph families are rather small scale (up to 64) nodes. Seeing how the proposed method performs on benchmark instances and comparing it with the algorithm seems quite important. Furthermore, it consolidates that this is a relevant algorithm in the context of neural solvers for the problem which provides support for the overall contribution.
>
> The scale aspect has been somewhat addressed in the rebuttal although not to a sufficiently diverse range of graph families. Then there's also some evidence that the model performs slightly worse than the algorithm in one of the problems in the instances that were already looked at. Even if I disregard the issue of where this model stands compared to other NNs for solving such problems, it's still not entirely convincing that this has learned to run the algorithm. The OOD results on other graph families are only shown for vertex cover but not on MSC. MSC OOD results would also help since those were the weaker numbers in the first table. Another interesting benchmark here would be to vary some of the generation parameters for the random graph models (edge density and so on) and test OOD there as the instances grow. If we are to just focus on whether the actual simulation of the algorithm happens (or improvement upon it anyway), then it's crucial to see plenty of diverse data points that support this conclusion, given we cannot promise this in theory, especially because we already see some instances were the model does not match the performance of the algorithm.Going beyond OOD evaluations, the premise of NAR is that such learned algorithms will aid in real world tasks where the data will be naturally high dimensional. However, the evaluation we see in table 4 is not compelling enough. Comparing with n2v and degree features is rather weak given the extensive research that exists on positional encodings for GNNs. At least a somewhat stronger baseline there (i.e., Laplacian eigenvectors or random walk features) might be good but ultimately we are checking whether this can succeed for node classification and there exist strong baselines for that task which are not being compared with. So it's not quite clear what exactly is being demonstrated with that experiment or how well it supports the overall story.
>
> I understand that I'm being a bit difficult here with the experiments but I would like to ensure that if a paper gets accepted for this category, it actually achieves something interesting and we're not just seeing an incomplete evaluation. Seeing results in a neural comb opt setting against other neural baselines is a useful proxy for me to see whether this method can consistently perform well.
>
> To summarize, I think more work needs to be done to get this over the acceptance threshold. Some of the results do look promising but getting a more complete set of experiments like I explained would help clearly outline where the contribution lies in this approach. I have increased my score but I don't think I can recommend acceptance at this point.

---

> ### Author Response · Authors · 2024-11-27
>
> We are grateful to Reviewer Y8qc for raising the score and acknowledging our additional experiments have improved the concerns raised. While we thank the reviewer for the follow-up response, we would like to politely and respectfully point out that:
> - Both our baseline selection and OOD evaluation settings closely follow previous NAR works. **A recent example is [2] (ICLR 2023 - notable top 25%)**, and we covered ~20 times larger sizes and more graph families than them.
> - Real-world experiments also follow [2], where node2vec was used as a baseline. We additionally compare with degree embedding.
>   - Node2vec is fundamentally different from positional encodings (PEs). Node2vec generates node embeddings that capture both structural and contextual patterns that are specifically helpful for downstream tasks. PEs are designed to capture absolute or relative positions, which are primarily useful for graph transformers (which operate on a fully-connected graph) to compensate for the loss of positional information.
>   - Node2vec has the advantage to be pretrained on the graph, while our model only performs inference on it.
> - (Wrt point 3) We agree with the reviewer that more comprehensive baseline comparison (other than General Point 3) from NCO literature will strengthen the paper’s contribution to NCO. However, we note that
>   - While MVC is explored in NCO (which we provided additional experiments on hard instances), MSC and MHS are less explored to have benchmarks. Many NCO models require careful modelling and reformulation to adapt to different CO problems, making it even more difficult.
>   - We consider contribution to NCO out of the scope of our paper, though we believe our method has potential to contribute to NCO (as explained in General Point 2), but we leave for future work.
>
> Once again, we are deeply grateful to Reviewer Y8qc for their time and feedback on our paper. Please don't hesitate if you have any other questions for us, and we are happy to address any of them.
>
> [2] Numeroso et al., Dual Algorithmic Reasoning, ICLR 2023 (notable top 25%).

---

> ### Comment · Reviewer_Y8qc · 2024-11-27
>
> - I do not dispute that the conventions followed are in line with previous work. That is a useful signal to some extent, but just because something was published at a certain standard before it doesn't mean that those standards cannot or should not be improved. I still have to judge this paper on its own merits while also taking context into account.
>
> - You compared with degree embeddings and with node2vec. The performance difference between those 2 is roughly within std in your tables. There has been extensive research on different kinds of PEs for GNNs. My usage of the term positional encoding is line with papers like this recent one [1]. I gave examples of exactly what I mean: when training GNNs eigenvectors and/or random walk features are among some of the options that can be used.  Is this applicable to your setting or have I misunderstood something? There are GAT + degree encoder results so I don't seen why GAT + random walk features is not a viable comparison. I am not asking you to benchmark all different kinds of PEs. I am saying that degree encodings are a weak baseline, n2v seems quite close to them, and other reasonable alternatives that are known to work better exist. There's extensive literature on this topic that goes beyond those, because different PEs have different expressive power and this can impact classification performance. Furthermore, if the ultimate goal is to do classification, I'm not sure why the comparison is not done against stronger baselines. Sure, the features may be better than n2v and degree encodings but why is that important? For this task there are probably much stronger models+PEs altogether, so I'm not sure what exactly are we learning from this experiment apart from the fact that the learned features are somewhat transferrable (although it is not clear why one would prefer to use them over something else). If the features improved the performance of several models compared to some of the best known PEs, then OK that could be an interesting contribution. But this experiment isn't nearly strong enough to make that kind of statement.
>
> - I appreciate that you added results for MVC and that's why I increased my score. Even though the model is not beating all the other baselines, this is still good OOD performance given that you trained in small BA graphs. That's definitely an encouraging signal.
>
> - You are claiming a general algorithm for different problems. I think it is fair to ask whether this methodology pans out on all those problems in an OOD setting or if it only works out for vertex cover. If it's only consistent for vertex cover, then it would be important to understand why, no? Size generalization seems to be fine for all problems so that is good. The evaluations do not have to be the same across the board for all the problems but having at least a couple of different graph family datasets with larger instances for each of the problems seems like a reasonable thing to ask to ensure that indeed the algorithm is consistently performed or whether the distribution of graphs matters.
>
>
> 1. Grötschla, Florian, Jiaqing Xie, and Roger Wattenhofer. "Benchmarking Positional Encodings for GNNs and Graph Transformers." arXiv preprint arXiv:2411.12732 (2024).

---

> ### Author Response · Authors · 2024-11-28
>
> We thank the reviewer for the engagement with our rebuttal. While we believe our previous response answers most of the questions, we now provide more details:
>
> - While we understand the reviewer’s point, we also appreciate the reviewer for finding the past accepted NAR work useful as a reference. Indeed, we follow past high-quality NAR publications accepted at ICLR when designing the evaluation settings.
> - We improved over the evaluation settings from [2] by additionally using degree embedding to contrast with our model’s vertex-cover embedding. This is because all three node classification tasks are (to some degree) related to predicting node influence (i.e. how influential a node is). It allows us to benchmark with our model’s embedding to show it is capturing more than degree information. Degree embedding in this context is a strong baseline because nodes with higher degrees are more likely to have high influence or be in vertex cover. Therefore, degree embedding is chosen because (i) it is closely related with prediction task (ii) helps to contrast with vertex-cover embeddings from our model. Both degree, vertex-cover, and n2v are capturing the notion of local “context”, which is not what PEs are designed for. We are well-aware of the current SOTA positional encoding literature. Positional encodings capture global (Laplacian) or relative (random walk) positional information of nodes in a graph, which can help GNNs to differentiate nodes with similar local neighborhoods for better expressivity (wrt. 1-WL test and locality) and help Graph Transformers to compensate for the loss of positional information (which is the more emphasized usage). Therefore, positional encodings are primarily helpful, designed and evaluated, in graph-level tasks (see [3][4][5][6] for recently published PE works). Given these reasons, we believe positional encodings are less relevant for our datasets, though technically they can be tested to validate our hypothesis, but now the rebuttal time for extra experiments has passed. Here, we are just hoping to clarify the rationale of evaluating against the degree embedding.
> - We thank the reviewer for acknowledging the additional experiments we performed on MVC which show good generalization ability and potential to NCO.
> - We are grateful to the reviewer for acknowledging the merit in size generalization of our model. We only performed ood graph family test for MVC because ood is well defined for ordinary graphs. For MSC and MHS, they are not graph problems. Problem instances are therefore generated using the bipartite representation, which is less straightforward in terms of ood graph family. An alternative is varying the random probability to generate bipartite graphs, but we are constrained by time now. However, we want to point out that we chose a diverse range of ood graphs for MVC (density, optimal cover size, etc.), where graphs come from different probabilistic models, presenting harder ood settings than varying the random probability within the same graph family.
>
> We appreciate the reviewer’s detailed feedback and engagement with our rebuttals. We have carefully considered all points raised and addressed as much as we can within the time limit of the rebuttal. We thank the reviewer again for their time spent on our paper!
>
> [3] Dwivedi et al., Graph neural networks with learnable structural and positional representations, ICLR 2022.
> [4] Lim & Robinson et al., Sign and basis invariant networks for spectral graph representation learning, ICLR 2023.
> [5] Huang & Lu et al., On the stability of expressive positional encodings for graphs, ICLR 2024.
> [6] Cantürk & Liu et al., Graph positional and structural encoder, ICML 2024.

---

### Author Response · Authors · 2024-11-21
**General Response**

We sincerely thank all the reviewers for their time and thoughtful feedback on our work. We are grateful to the reviewers for their positive recognition of our method as both **interesting** and **effective** (Y8qc, YZYW, KsRr, EkCx), our experimental results as **comprehensive** and **favorable** (Y8qc, YZYW, KsRr), our theoretical justification as **helpful** (YZYW), and our writing as **detailed**, **clear** and **easy to follow** (KsRr, EkCx). Additionally, we note that the majority scores for Soundness, Presentation, and Contribution across all three axes were 3: good, reflecting a strong overall assessment of our work.

We deeply value the reviewers' concerns and have given them serious consideration. Below, we address the common issues raised by the reviewers. Additionally, we have made substantial revisions to the manuscript (marked in blue) to incorporate these clarifications and emphasize our contributions. We sincerely thank the reviewers for their constructive feedback, which has helped us refine our work. We hope these revisions effectively address the reviewers’ concerns and provide a clearer understanding of the novelty and significance of our approach.

---

> ### Author Response · Authors · 2024-11-21
> **Point 1. Our primary focus is on Neural Algorithmic Reasoning (NAR)**
>
> Reviewers Y8qc, YZYW, KsRr raised concerns about the lack of relevant baselines. We believe this may stem from a misunderstanding of the primary focus of our paper, which is rooted in Neural Algorithmic Reasoning (NAR) rather than Neural Combinatorial Optimization (NCO). Our evaluation closely follows the settings established in prior NAR works, including the choice of baselines and datasets. To address this concern, we have provided additional context on NAR, which is now reflected in the Introduction and Related Works sections of the revised manuscript.
>
> - **What is NAR**: NAR [1] combines the power of deep learning with the structured logic of classical algorithms to create models capable of simulating algorithmic processes. The central idea is to train neural networks to replicate the step-by-step operations of traditional algorithms, such as Bellman-Ford for the shortest-path problem. This approach equips models with algorithmic reasoning capabilities, making them well-suited for tasks requiring structured reasoning or domains where problems can be mapped to algorithms. For instance, a model pre-trained with knowledge of the Bellman-Ford algorithm can be applied to tackle real-world routing and transportation problems. NAR takes advantage of neural networks' flexibility to work directly with real-world data, removing the need for feature engineering to transform problems into the abstract formats required by traditional algorithms. This helps to bridge the gap between classical methods and real-world applications.
>
> - **Problem**: Most NAR research has focused on algorithms for polynomial-time-solvable problems, such as the 30 classic algorithms (sorting, search, graph) studied in CLRS-30 [2]. This creates a significant gap when addressing real-world problems, many of which are inherently NP-hard. This gap is critical to address, as the motivation behind NAR is to enable the transfer of algorithmic knowledge to tackle complex, real-world datasets effectively.
>
> - **Related work**: The most closely related work that attempts to extend supervised NAR to NP-hard problems is [3]. This approach involves pretraining a GNN on algorithms for polynomial-time-solvable problems (e.g., Prim’s algorithm for MST) and applying transfer learning to NP-hard problems (e.g., TSP). However, this method is not general, as it relies on carefully selecting a specific algorithm for pretraining, requiring the identification of a meaningful link between the polynomial-time problem and the target NP-hard problem. This limitation underscores the need for a more generalizable and effective framework.
>
> - **We propose a general NAR framework to learn algorithms for NP-hard problems**: Our framework significantly expands NAR's applicability to NP-hard domains. A key challenge in applying NAR to NP-hard problems has been the lack of intermediate ground-truth labels to guide learning, as noted in [2]. We address this by leveraging a primal-dual approximation algorithm as a foundational framework and using optimal solutions from smaller instances to enhance the model to produce high-quality solutions. Furthermore, unlike prior works that focus on accurately replicating the algorithmic behavior, our framework is explicitly designed to outperform the algorithm it is trained on.
>
> - **We closely follow and cover the standard evaluation settings in NAR literature**: The central goal of NAR is to assess the generality and robustness of algorithmically informed models. We closely follow the evaluation settings from previous works: (i) OOD generalization: It is typically evaluated by training on small graphs of size 16 and testing on increasingly larger graph sizes (16, 32, 64) as well as on out-of-distribution graph families. (ii) Baseline comparison: The trained model is compared with non-algorithmically informed baselines. The intent is to demonstrate that algorithmic knowledge is effective in enabling structured reasoning and aiding generalization. Additionally, we compare with models that do not incorporate optimal solutions to show that these signals allow the model to surpass the algorithm’s performance. (iii) Real-world experiments: This is evaluated by pretraining the model on synthetic algorithmic data and testing its transferability to real-world problems, such as standard node classification tasks.
>
> For reference, **a recent NAR example [4] (ICLR 2023 notable 25%)** that clearly demonstrates all three evaluation settings.

---

> ### Author Response · Authors · 2024-11-21
> **Point 2. Comparison with Neural Combinatorial Optimization (NCO)**
>
> We acknowledge that our work overlaps with NCO in its application to NP-hard combinatorial optimization problems. To further address the reviewers’ concerns, we emphasize that NAR and NCO are fundamentally distinct in their objectives and methodologies. We also added a discussion of our work on NCO in the Related Works and Appendix H & I in the revised manuscript.
>
> **Difference with NCO**: NCO focuses on solving CO problems by using neural networks to learn task-specific heuristics or optimize solutions end-to-end. Its primary goal is to find high-quality solutions to NP-hard problems efficiently and directly, which is performance-driven. In contrast, NAR is centered on designing neural architectures that emulate the reasoning processes of classical algorithms. Unlike NCO, which aims to directly construct high-quality solutions for optimization problems, NAR emphasizes capturing and generalizing the underlying reasoning processes of algorithms. The goals of NAR are threefold:
> - *Reasoning ability*: We demonstrate that GNNs can effectively simulate primal-dual algorithms designed for approximating NP-hard problems. By incorporating optimal solutions during training, GNNs can even surpass the performance of these algorithms (Table 1).
> - *Generalization*: Our results highlight that embedding algorithmic knowledge significantly enhances generalization. GNNs trained on primal-dual algorithms generalize well to larger problem instances (Table 1) and different graph distributions (Table 2).
> - *Embedding algorithmic knowledge for real-world data*: NAR employs an encoder-processor-decoder framework. We show that a GNN processor pretrained with primal-dual algorithm, combined with a new encoder, can directly process real-world datasets without requiring feature engineering to conform to rigid input formats of traditional algorithms. This produces embeddings that effectively capture algorithmic knowledge, improving performance on node-classification tasks in real-world applications (Table 4).
>
> We also appreciate the reviewers for highlighting the relevance of our work to NCO, which broadens the applicability of our method beyond NAR. While NCO is not the primary focus of our work, we recognize its importance and have elaborated on our position within the NCO literature to provide a clearer context.
>
> **Relevance of our work to NCO**: Our work also contributes to NCO by studying an underexplored area of NCO, proposing an algorithmically informed GNN that effectively addresses critical data efficiency and generalization challenges. Most GNN-based supervised learning methods for NCO learn task-specific heuristics or optimize solutions in an end-to-end manner [5][6][7][8]. These end-to-end approaches rely exclusively on supervision signals derived from optimal solutions, which are computationally expensive to obtain for hard instances. Furthermore, dependence on such labels can limit generalization [9]. In contrast, our method trains on intermediate supervision signals obtained efficiently from a polynomial-time approximation algorithm. The embedding of algorithmic knowledge also demonstrates strong generalization. Additionally, supervising with optimal solutions derived from small problem instances enables our model to outperform the approximation algorithm and generalize effectively to larger problem sizes, addressing key data efficiency and generalization bottlenecks.

---

> > ### Author Response · Authors · 2024-11-21
> > **Point 3. Comparison with NCO baselines**
> >
> > To directly address reviewers’ concerns, we also include extra NCO baselines in our experiments to validate the strengths of our approach. Based on Reviewer Y8qc’s suggestion, we have conducted additional experiments on the RB benchmark graphs (well-known hard instances for MVC), and compared our method with several NCO baselines. Given the supervised nature of our approach, obtaining optimal solutions for RB200/500 graphs is computationally challenging. Therefore, we tested the generalization of our model, trained on Barabasi-Albert graphs of size 16 (using intermediate supervision from the primal-dual algorithm and optimal solutions), on the larger RB200/500 graphs. We also included results from EGN [10] and Meta-EGN [11], two powerful NCO baselines for these benchmarks, as well as two algorithms (primal-dual approximation algorithm, greedy algorithm) and Gurobi.
> >
> > Table R1: Approximation ratio of solutions compared with optimal solutions for MVC (lower is better).
> >
> > | Method               | RB200            | RB500            |
> > |----------------------|------------------|------------------|
> > | EGN                 | 1.031 ± 0.004    | 1.021 ± 0.002    |
> > | Meta-EGN            | 1.028 ± 0.005    | 1.016 ± 0.002    |
> > | PDGNN (ours)        | 1.029 ± 0.005    | 1.020 ± 0.004    |
> > | Primal-dual         | 1.058 ± 0.003    | 1.056 ± 0.004    |
> > | Greedy              | 1.124 ± 0.002    | 1.062 ± 0.005    |
> > | Gurobi9.5 (≤ 1.00s) | 1.011 ± 0.003    | 1.019 ± 0.003    |
> > | Gurobi9.5 (≤ 2.00s) | 1.008 ± 0.002    | 1.019 ± 0.003    |
> >
> > From Table R1, we see PDGNN outperforms EGN and is competitive with its improved variant, Meta-EGN. Notably, EGN and Meta-EGN are trained directly on 4000 RB200/500 graphs, while our method was trained solely on 1000 B-A graphs with just 16 nodes and was tested out-of-distribution. This highlights the data efficiency and strong generalization of PDGNN.
> >
> > We note that both EGN and Meta-EGN are unsupervised methods, while our method adopts a supervised approach. We acknowledge that a fairer comparison would involve supervised baselines for NCO. However, recent focus in NCO has shifted toward unsupervised learning due to the difficulty in acquiring labels. Existing supervised baselines often rely on outdated codebases that are challenging to adapt within the short rebuttal period. Additionally, obtaining labels for large graphs such as RB200/500 is computationally prohibitive. We hope to seek reviewers’ understanding given these constraints and the fact that our paper primarily focuses on NAR rather than NCO. We believe the baseline comparison underscores the potential of our approach for supervised NCO, demonstrating the effectiveness of an algorithmically informed GNN with efficiently obtainable labels and strong generalization capabilities.

---

> > > ### Author Response · Authors · 2024-11-21
> > > **References**
> > >
> > > [1] Veličković and Blundell, Neural Algorithmic Reasoning, Patterns, 2021.
> > >
> > > [2] Veličković et al., The CLRS Algorithmic Reasoning Benchmark, ICML 2022.
> > >
> > > [3] Georgiev et al., Neural Algorithmic Reasoning for Combinatorial Optimization, LoG 2023.
> > >
> > > [4] Numeroso et al., Dual Algorithmic Reasoning, ICLR 2023.
> > >
> > > [5] Joshi et al., An Efficient Graph Convolutional Network Technique for the Travelling Salesman Problem, 2019.
> > >
> > > [6] Li et al., Combinatorial Optimization with Graph Convolutional Networks and Guided Tree Search, NeurIPS 2018.
> > >
> > > [7] Gasse et al., Exact Combinatorial Optimization with Graph Convolutional Neural Networks, NeurIPS 2019.
> > >
> > > [8] Fu el al., Generalize a small pre-trained model to arbitrarily large tsp instances, AAAI 2021.
> > >
> > > [9] Joshi el al., Learning the Travelling Salesperson Problem Requires Rethinking Generalization, Constraints 2022.
> > >
> > > [10] Karalias & Loukas, Erdos Goes Neural: an Unsupervised Learning ˝ Framework for Combinatorial Optimization on Graphs, NeurIPS 2020.
> > >
> > > [11] Wang & Li, Unsupervised Learning for Combinatorial Optimization Needs Meta Learning, ICLR 2023.

---

### Meta-Review · Area_Chair_Eurd · 2024-12-17

**Metareview:**

This paper proposes a GNN framework using primal-dual approximation for NP-hard combinatorial optimization, aiming to enhance neural algorithmic reasoning. It shows potential in surpassing base algorithm performance and has practical applications. However, it has notable weaknesses. The experimental comparisons lack comprehensiveness, especially regarding out-of-distribution performance and relevant baselines. The approximation guarantee is not fully convincing, and the work's position relative to related methods is not as clear as it should be. Overall, these issues lead to the paper's rejection as the evidence and clarity of its contributions do not meet the required standards.

The strengths of the paper include its novel approach and theoretical justifications. The authors also made efforts during the rebuttal to address concerns. But the remaining weaknesses in the experimental design and theoretical underpinnings are significant enough to outweigh the strengths. The lack of a more conclusive demonstration of its superiority and generalizability compared to existing methods makes it unready for acceptance at this stage.

**Additional Comments On Reviewer Discussion:**

Reviewers raised various concerns during the rebuttal, such as the need for more diverse baselines and better-defined evaluations of out-of-distribution performance. The authors responded by conducting additional experiments and providing more detailed explanations. However, in the final assessment, these efforts did not fully address the reviewers' concerns. The paper's contribution in the context of existing literature remains somewhat unclear, and the experimental results do not convincingly demonstrate its superiority and generalizability.

---

### Decision · Program_Chairs · 2025-01-22

Reject